# Application of a physically-based model to forecast shallow landslides occurrence at regional scale

Teresa Salvatici[1,] Veronica Tofani[1], Guglielmo Rossi[1], Michele D'Ambrosio[1,] Carlo Tacconi Stefanelli[1], Elena Benedetta Masi[1], Ascanio Rosi[1], Veronica Pazzi[1], Pietro Vannoci[1], Miriana Petrolo[1], Filippo Catani[1], Sara Ratto[2], Hervè Stevenin[2] and Nicola Casagli[1]

[1]Department of Earth Sciences, University of Firenze, Firenze, 50121, Italy
[2]Centro funzionale, Regione Autonoma Valle d'Aosta, Aosta, 11100, Italy

*Correspondence to*: Michele D'Ambrosio (michele.dambrosio@unifi.it)

Abstract.

In this work, we apply a physically-based model, namely the HIRESSS (High REsolution Stability Simulator) model, to forecast the occurrence of shallow landslides at regional scale. HIRESSS is a physically based distributed slope stability simulator for analysing shallow landslide triggering conditions during a rainfall event. The software is made of two parts: hydrological and geotechnical. The hydrological model is based on an analytical solution of an approximated form of the Richards equation while the geotechnical stability model is based on an infinite slope model that takes into account the unsaturated soil condition. The test area is a portion of the Valle d'Aosta region, located in North-West Alpine mountain chain. The geomorphology of the region is characterized by steep slopes with elevations ranging from 400 m a.s.l of Dora Baltea's river floodplain to 4810 m a.s.l. of Mont Blanc. In the study area, the mean annual precipitation is about 800-900 mm. These features lead to the territory to be very prone to landslides, mainly shallow rapid landslides and rock falls. In order to apply the model and to increase its reliability, an in-depth study of the geotechnical and hydrological properties of hillslopes controlling shallow landslides formation was conducted. In particular, two campaigns of on site measurements and laboratory experiments were performed with 12 survey points. The data collected contributes to generate input map of parameters for HIRESSS model. In order to consider the effect of vegetation on slope stability, the soil reinforcement due to the presence of roots has been also taken into account based on vegetation maps and literature values of root cohesion. The model was applied in back analysis on two past events that have affected Valle d'Aosta region between 2008 and 2009, triggering several fast shallow landslides. The validation of the results, carried out using a database of past landslides, has provided good results and a good prediction accuracy of the HIRESSS model both from temporal and spatial point of view.

## 1 Introduction

Landslide prediction at regional scale can be performed following two approaches: a) rainfall thresholds based on statistical analysis of rainfall and landslides and b) physically-based deterministic models. While the first approach is currently extensively used at regional scale (Aleotti, 2004; Cannon et al., 2011; Martelloni et al., 2012; Rosi et al., 2012; Lagomarsino et al., 2013), the latter is more frequently applied at slope or catchment scale (Dietrich and Montgomery 1998; Pack et al. 2001; Baum et al. 2002, 2010; Lu and Godt 2008; Simoni et al. 2008; Ren et al. 2010; Arnone et al. 2011; Salciarini et al., 2012; Park et al., 2013; Rossi et al. 2013; Salciarini et al. 2017). The poor knowledge of hydrological and geotechnical parameters spatial distribution, caused by the extreme heterogeneity and inherent variability of soil at large scale (Mercogliano et al., 2013; Tofani et al., 2017), mainly avoid the physically-based model application at regional scale. On the other hand, physically-based models allow to predict spatially and

temporally the occurrence of landslides with high accuracy producing accurate hazard maps that can be of help for
landslide risk assessment and management.
In this work, we apply the physically based model, named HIRESSS (Rossi et al., 2013) in Eastern part of Valle
d'Aosta region (Italy), in North-West Alpine mountain chain in order to test the capacity of the model to forecast the
occurrence of shallow landslides at regional scale. In particular, the objectives of the work are: i) to properly
characterise the geotechnical and hydrological parameters of the soil to feed the HIRESSS model and to spatialize this
punctual information in order to have spatially-continuous maps of the model input data ii) to test the HIRESSS code
for two selected rainfall events that have triggered several shallow landslides and to validate the model results.
HIRESSS is a physically based distributed slope stability simulator for analysing shallow landslide triggering
conditions in real time and in large areas using parallel computational techniques. In the area selected, an in-depth study
of the geotechnical and hydrological properties of hillslopes controlling shallow landslides formation was conducted,
performing two campaigns (12 survey points) of in-situ measurements and laboratory tests. Furthermore, the HIRESSS
model has been modified to take into account the effect of the root reinforcement to the stability of slopes based on
plant species distribution and literature values of root cohesion.

## 53  2 Study area and rainfall events

The study area, called alert Zone B by the regional civil protection authorities, is located in eastern part of Valle
d'Aosta region, in North-West Alpine mountain chain (Fig. 1). The area is characterized by three main valleys:
Champorcher valley, Gressoney or Lys valley, and Ayas valley. The first is located on the right side of Dora Baltea
water catchment, and represents the southern part of the study area. The second and third valleys show N-S orientation,
and they are delimited to north by Monte Rosa massif (4527 m a.s.l) and to south by Dora Baltea river.
From a geological point of view, the Valle d'Aosta is located NW with respect to the Insubrica Line, in particular, there
are three systems of Europa chain: the Austroalpino, the Pennidiche and the Elvetico-Ultraelevato systems (De Giusti,
2004). Fig. 2 shows the lithological map of the study area obtained by reclassifying the geological units according to 11
lithological groups: landslides, calcareous schist, alluvial deposits, glacial deposits, colluvial deposits, glacier, granites,
mica schists, green stone, black schists and serpentinites. In detail in the study area the main lithologies outcropping are
metamorphic and intrusive rocks, in particular granites, metagranites, schists and serpentinite.
The geomorphology of the region is characterized by steep slopes and valleys shaped by glaciers. The glacial modelling
is shown in the U-shaped of Lys and Ayas valleys, and the erosive depositional forms found in the Ayas valley. The
three valleys' watercourses, the Lys creek, the Evançon creek, and the Dora Baltea river, contributed to the glacial
deposits modelling with the formation of alluvial fans. The climate of the region is characterized by high variability
strongly influenced by altitude (ranging from 400 m a.s.l of Dora Baltea's river floodplain to 4810 m a.s.l. of Mont
Blanc), with a continental climate in the valleys floor and an Alpin climate at high altitudes.
The slope steepness, together with mean annual precipitation of 800-900 mm are the main landslide triggering factors.
These features lead the study area to be prone to landsliding, in particular rock falls, deep seated gravitational slope
deformations (DSGSD), rocks avalanches, debris avalanches, debris flows, and debris slides (Catasto dei Dissesti
Regionale – form Val d'Aosta Regional Authorities). In this work we model the triggering conditions of shallow
landslides, i.e. soil slips and translational slides and we do not take into account the other types of movement.
The HIRESSS model simulated two past events, one in 2008 and one in 2009, and the validation of the model
performance was carried out comparing the results with the landslide regional database.
In particular:
•    24 - 31 May 2008: on 28 and 29 May 2008 intense and persistent rainfall was recorded across the Valle d'Aosta

region with a total precipitation in the study area of about 250 mm causing flooding, debris flows and rockfalls.

•    25 - 28 April 2009: from 26 April to 28 April 2009 heavy rainfall affected the south-eastern part of the Valle

d'Aosta region, with the highest precipitation recorded at the Lillianes Granges station of about 268 mm. This

precipitation triggered several landslides.


## 3 Methodology

### 3.1 HIRESSS description

The physically-based distributed slope stability simulator HIRESSS (Rossi et al., 2013) is a model developed to analyse
shallow landslide triggering conditions on large scale at high spatial and temporal resolution using parallel calculation
method. Two parts compose the model: hydrological and geotechnical (Rossi et al., 2013). The hydrological part is
based on a dynamical input of the rainfall data which are used to calculate the pressure head and provide it to the
geotechnical stability model. The hydrological model is initiated as a modelled form of hydraulic diffusivity, using an
analytical solution of an approximated form of the Richards equation under the wet condition (Richards, 1931). The
equation solution allows us to calculate the pressure head variation ($h$), depending on time ($t$) and depth of the soil ($Z$).
The solutions are obtained by imposing some boundary conditions as described by Rossi et al. (2013).
The geotechnical stability model is based on an infinite slope stability model. The model considers the effect of matric
suction in unsaturated soils, taking into account the increase in strength and cohesion. The stability of slope at different
depths (Z values) is computed since the hydrological model calculates the pressure head at different depths. The
variation of soil mass caused by water infiltration on partially saturated soil is also modelled. The original FS equations
(Rossi et al., 2013) were modified taking into account the effect of root reinforcement ($c_r$) as an increase of soil
cohesion ($c'$) according to the Eq. 1:
$c_{tot} = c' + c_r$

(1)

Regarding the geotechnical influence of roots on the soil strength, roots seem to affect the cohesion parameter only,
while the friction angle would be poorly or not at all interested by reinforcement (Waldron and Dakessian, 1981; Gray
and Ohashi 1983; Operstein and Frydaman, 2000; Giadrossich et al., 2010). Therefore, is necessary to consider the root
cohesion in calculating FS and consequently in applying HIRESSS model.
The root reinforcement (or root cohesion) can be considered equal to (Eq. 2):
$c_r = kT_r(A_r/A)$

(2)

where Tr is the root failure strength (tensile, frictional, or compressive) of roots per unit area of soil, Ar/A the root area
ratio (proportion of area occupied by roots per unit area of soil), k a coefficient dependent on the effective soil friction
angle and the orientation of roots. The measure of cr varies with vegetal species, within a single species depends on
how plants respond to environmental characteristics and fluctuations.

The new equation of FS at unsaturated conditions is therefore (Eq. 3):
$$FS = \frac{\tan\varphi}{\tan\alpha} + \frac{c_{tot}}{\gamma_d y \sin\alpha} + \frac{\gamma_w h \tan\varphi\left\{\left[1+\left(h_b^{-1}|h|\right)^{\lambda+1}\right]^{\frac{\lambda}{\lambda+1}}-1\right\}}{\gamma_d y \sin\alpha}$$

(3)

where $\varphi$ is the friction angle, $\alpha$ is the slope angle, $\gamma_d$ is the dry soil unit weight, $y$ is the depth, $\gamma_w$ is the water unit
weight, $h$ is the pressure head, $h_b$ is the bubbling pressure, and $\lambda$ is the pore size index distribution. In saturated
condition the equation of FS (Rossi et al., 2013) becomes (Eq. 4):
$$FS = \frac{\tan\varphi}{\tan\alpha} + \frac{c_{tot}}{(\gamma_d(y-h)+\gamma_{sat}h)\sin\alpha} - \frac{\gamma_w h \tan\varphi}{(\gamma_d(y-h)+\gamma_{sat}h)\tan\alpha}$$

(4)

where $\gamma_{sat}$ is the saturated soil unit weight.
One of the major problems, associated with the deterministic approach employed on a large scale, is the uncertainty of
the static input parameters or geotechnical parameters of the soil. The method used for the estimation of parameters
spatial variability is the Monte Carlo Simulation. The Monte Carlo simulation achieves a probability distribution of
input parameters providing results in terms of slope failure probability (Thiery et al. 2017). The developed software
uses the computational power offered by multicore and multiprocessor hardware, from modern workstations to
supercomputing facilities (HPC), to achieve the simulation in reasonable runtimes, compatible with civil protection real
time monitoring (Rossi et al. 2013). The HIRESSS model loads spatially distributed data arranged as 12 input raster
maps and the maps of rainfall intensity. These input raster maps are: slope gradient; effective cohesion ($c'$); root
cohesion ($c_r$); friction angle ($\varphi'$); dry unit weight ($\gamma_d$); soil thickness; hydraulic conductivity ($k_s$); initial soil saturation
($S$); pore size index ($l$); bubbling pressure ($h_s$); effective porosity ($n$); and residual water content ($q_r$). and rainfall
intensity.

### 3.2 HIRESSS input data preparation

The input parameters can be divided in two classes: the static data and the dynamical data. Static data are geotechnical
and morphological parameters while the dynamical data is represented by the hourly rainfall intensity. Static data are
read only once at the beginning of the simulation while dynamical inputs are continuously updated.
The HIRESSS input are in raster, therefore point data and parameters have to be adequately spatially distributed. In this
application the spatial resolution was 10 m.
*Static data*
The slope gradient was calculated from the DEM (Digital Elevation Model- 2006). Effective cohesion, friction angle,
hydraulic conductivity, effective porosity and dry unit weight, were obtained, spatializing according to lithology, the
soil punctual parameters derived from the in situ and laboratory geotechnical tests and analysis.
In particular, the properties of slope deposits were determined by in situ and laboratory measurements (Bicocchi et al.,
2016; Tofani et al., 2017) at 12 survey points. To carry out the in situ tests the survey points were selected following
these characteristics: i) physiography, ii) landslides occurrence, and iii) geo-lithology (Fig. 2). Regarding the first point,
a high-resolution DEM (from Val d'Aosta Regional Authorities) together with a careful first surveys were used to
locate the most suitable slopes. The surveys took place in two sessions, the first one in August 2016, and the second one
in September 2016. The following analyses were conducted:
•    registration of geographical position using a GPS and photographic documentation of the site characteristics

(morphology and vegetation);

•    in situ measurement of saturated hydraulic conductivity ($k_s$) by means of the constant-head well permeameter

Amoozemeter;

•    sampling of an aliquot (~2 kg each) of the material for laboratory tests, including grain size distributions, index

properties, Atterberg limits and direct shear tests.

The permeability in-situ measurements and the soil samplings were made at depth ranging from 0.4 to 0.6 m below the
ground level. The evaluation of the $k_s$ (saturated hydraulic conductivity or permeability) was made with the
*Amoozemeter* permeameter (Amoozegar, 1989). The measurement was obtained by observing the amount of water
required to maintain a constant volume of water into the hole. In situ measurements are then applied into the Glover
solution (Amoozegar, 1989). which calculates the saturated permeability of the soils. The $k_s$ is a very useful parameter
not only for slope stability modelling but also for many other hydrological problems (groundwater, surface water runoff
and sub-surface, flow calculation of water courses).
In addition, the in situ collected samples were examined in the laboratory to define a wide range of parameters to
characterize more extensively the deposits. In particular, the following tests were performed in order to classify the
analysed soils:
•    grain size distribution (determination of granulometric curve for sieving and settling following ASTM

recommendations), and classification of soils (according to AGI and USCS classification, Wagner, 1957);

•    determination of the main index properties (porosity, relationships of phases, natural water content $w_n$, natural

and dry unit weight $\gamma$ and $\gamma_d$) following the ASTM recommendations;

•    determination of Atterberg limits (liquid limit LL, plastic limit PL, and plasticity index PI);
•    direct shear test on selected samples.
Soil thickness was calculated by the GIST model (Catani et al., 2010; Del Soldato et al, 2016). Soil characteristic curves
parameters (pore size index, bubbling pressure, and residual water content) were derived from literature values (Rawls
et al., 1982).
Root cohesion variations in the area (at the soil depth chosen for the physical modelling with HIRESSS) were obtained
firstly, identifying the plant species and determining their distribution from *in situ* observations and vegetational maps
(Carta delle serie di vegetazione d'Italia, Italian Ministry of the Environment and Protection of Land and Sea). Then,
the measure of cohesion due to the presence of roots was assigned to each subarea according to the dominant plant
species and literature root cohesion for that species (Bischetti, 2009; Burylo et al., 2010; Vergani et el., 2013) that were
calculated considering the Fiber Bundle Model (Pollen et al., 2004). The measure of $c_r$ varies with vegetal species,
within a single species depends on how plants respond to environmental characteristics and fluctuations, so map of root
cohesion variations obtained as mentioned is a simplification of reality. This is a necessary simplification as the known
methods to evaluate root cohesion variations are not suitable for wide areas and acceptable measurement times.
The last static input data, in this case of study, is the exposure rock mask. This was defined considering the lithological
and land use maps, so that HIRESSS model avoided the simulation on steep rock slopes areas.
The geotechnical properties and root cohesion of the soils have been spatialized with respect to a lithological
classification.
For each lithological class and plant species the mean value has been selected in order to obtain the HIRESSS input
raster parameters.

*Dynamic data*
In the study area, the rainfall hourly data from 27 pluviometers were available, therefore it was necessary to spatially
distribute them to generate 10x10 m cell size input raster to ensure the correct program operation. The rainfall data were
elaborated applying the Thiessen's polygon methodology (Rhynsburger, 1973) modified to take into account the
elevation. Thiessen's polygon methodology, in fact, allows us to divide a planar space in some regions, and to assign the
regions to the nearest point feature. This approach defines an area around a point, where every location is nearer to this
point than to all the others. Thiessen's polygon methodology do not consider the morphology of the area, so the alert
Zone B was divided in three catchment areas and the polygons were calculated for each rain gauges considering the
reference catchment basin (Fig. 3).
**4 Results**
The results of the geotechnical and hydrological characterization of the soils of the 12 survey points are shown in Table
1 for all survey sites.
The results of granulometric tests shown that the analysed soils are predominantly sands with silty gravel (Fig. 4 and
Table 1). Regarding the index properties, the natural soil water content values were predominantly about 20% by
weight, with a maximum and minimum values of 5.1% and 26.2%, respectively. These values reflect their different
ability to hold water in their voids. The measured natural unit weight ($\gamma$) was variable between 15.3 kN/m$^3$ and 21.7
kN/m$^3$, depending not only on the different grain size distribution but also by different thickening and consolidation
states. Regarding saturated unit weight ($\gamma_{sat}$) the measured values range between 18.2 kN/m$^3$ and 21.5 kN/m$^3$ (Table 1).
The Atterberg limits (LL and PL) were measured on samples with a sufficient passing fraction (> 30% by weight)
through 40 ASTM (0.425 mm) sieve. For sandy prevalent samples, LL values are predominantly around 40% of water
content (% by weight), while the PL is around 30% (Table 1).
The effective friction angle varies between a minimum of 25.6° and a maximum of 34.3°, while the effective cohesion
ranges from a minimum of 0.0 kPa to a maximum of 9.3 kPa. Consistent with the presence of sandy soils, the saturated
permeability values were around a medium-high value of 10$^{-6}$ m/s. The minimum and maximum values were found
between $1.36 \cdot 10^{-7}$ m/s and $1.54 \cdot 10^{-5}$ m/s. Considering the poor variability of samples, the permeability values were
relatively homogeneous and in accordance with the values reported in the literature (Table 1).
The additional cohesion induced by roots assumes different values not only depending on plant species and
environmental characteristics, but also on depth of soil, as roots diameter and density vary with latter. Because of such
evidence, studies on roots cohesion of different species report values as function of depth of soil. In the area of the case
study, soils have thinner thickness than those ones in which these studies are carried out. In such thin soils, root systems
organize their growth depending on available space not reaching the same depth of roots of thick soils. Consequently, in
this context root cohesion of species at the different depth is dissimilar related to literature values. Considering this,
map for variation of root cohesion is processed taking for each species the minimum cohesion (among those specified
for each species at the different depth) reported in literature. By doing this, contribution of vegetation to stability of
slopes is considered in FS calculate and at the same time, it is avoided an overestimate of root cohesion.
In the area, root cohesion defined as mentioned above ranges from a minimum of 0.0 kPa (mainly in the outcrop area)
to maximum of 8.9 kPa (area occupied by mountain maple on the left bank of river Dora Baltea).
In Table 2, the mean values of each input parameters respect to lithological class were reported.
The pore size index, bubbling pressure and residual water content are constant in whole area of: 0,322 (-); 0,1466 m and
0,041 (-), respectively.
The distributed soil parameters maps are shown in Fig. 5.
The results of rainfall data elaborated using Thiessen's polygon methodology are 192 and 96 rainfall hourly maps for
the 2008 and 2009 event, respectively. In Fig. 6 are reported the cumulative maps of each event.
All these data have been inserted in the HIRESSS model to obtain day-by-day maps of landslide occurrence probability.
The main characteristics of simulation are showed in Table 3. Before to discuss the model results is necessary to check
false positive for both the simulated events, the first day of simulation, characterized by the absence of rainfall, was
analysed. The results showed that those pixels with a high landslide occurrence probability are unstable because of
morphometric reasons, predominantly high slope angles. To remove these false positive, a numeric mask was applied.
Using the GIS software commands, it was possible to calculate the number of pixels of the first simulation day with a
trigger probability value greater than 80% and delete them (Fig. 7). The mask was then applied to the rest of landslide
occurrence probability maps. The resulting maps for each days of the simulated events are shown in the Fig. 8 and Fig.

244 9.

**5    Discussion**
A back analysis was carried out to evaluate the model performance from a temporal and spatial point of view. To
perform a solid validation is necessary to have information on spatial and temporal location of landslides. In particular,
the time of occurrence is very rarely known with hourly precision, and usually landslides are related to a rainstorm,
without any more precise information on time of occurrence (Rossi et al., 2013). Concerning the spatial landslides
locations, in many cases they are included in the database only as points without any information on the area involved.
In our database, provided by the local authorities, landslides are points with information on the day of occurrence.
In general, for both events temporal validation shows that the daily highest probability of occurrence, computed by
HIRESSS, correspond with the days with real landslide occurrence and with the most intense precipitation.
The results of the first simulated event (24 - 31 May 2008) are shown in Fig. 8. The failure probability in the whole area
is negligible for the first four days (from 24 to 27 May 2008) (Fig. 8a). The rainfall intensity increased since 27 May,
reaching the highest value on 29 May, when the precipitation value was around 100 mm in the eastern sector of study
area.
The HIRESSS model well simulate this passage: the 28 May and 29 May 2008 landslide occurrence probability maps
show a considerable increase of the probability of failure with maximum values around 90% at the East of alert Zone B
(Fig. 8 b, c). In the following days rainfall intensity decreases, and also the probability slowly decreases, being anyway
still high on 30 May 2008. Landslides reported in the database are dated 30 May and 31 May 2008 (Fig. 8d).
Concerning the second event (25 - 28 April 2009) landslide occurrence probability is negligible for the first two days
(25 and 26 April 2009) in the whole area (Fig. 9 a, b), because of the low rainfall intensity. From 27 April 2009 rainfalls
become more intense, especially in the southeast sector of the region, where the cumulated rainfall average was about
151 mm. This event led to many landslides triggered during these days (as reported in the database). Also the
probability maps show high values during these days (Fig. 9 c, d).
In Table 4 the results over 75% of slope failure probability for both events are highlighted and confirm the correct
temporal occurrence of landslides.
The temporal validation was also carried out considering daily cumulative rainfall compared to the landslide failure
probability. In particular, a median of landslide occurrence probability was calculated for four pluviometric areas
identified by Thiessen's polygons methodology, modified according to limits of river basins, both for the event of May
2008 and for the April 2009 event (Fig. 10 a, b). As it could be expected, the results show that when the highest rainfall
intensity is measured, the highest probability of occurrence is computed for the all areas and for both events.

Spatial validation was performed following a pixel by pixel method: this method is the most complex since it consists in
comparing the probability of instability of each pixel with the pixels involved in the actual event that occurred. This
validation implies a great deal of uncertainty in the results since the reports of landslide events may have errors on the
precise spatial location and on the size of the phenomenon. To overcome this problem and taking into account probable
errors caused by the actual spatial location in the database, an area of 1 km$^2$ (called influence area) around the point of
the landslide were considered in the validation analysis. Inside the influence area, pixels that have the 75% of
probability of failure were considered instable.
Figure 11 shows an example of landslide event occurred in the Arnad municipality on 30 May 2008. The model
computes a low failure probability on 24 May 2008 and an increase of probability on 30 May 2008. In Fig. 11a and b it
is possible to note that inside the red circle the red and yellow area increase on 30 May with respect to 24 May. In this
case, the model is able to identify correctly such movement. To better highlight this validation, Figure 10c shows the
number of pixels above 75% of probability calculated by the model, within the circular area of about 1 km$^2$ around the
all landslides occurred during the event of 2008. For some of the reported landslide events, the number of pixels above
75% increases on 30 May,2008, only in case of the Champdepraz and Montjovet 2 events the probability does not
increase. This may be caused by the low precision of location of the reported landslide, and maybe because some of the
real landslides reported are other types of movements (rockfalls, rotational slides) that cannot simulated by the
HIRESSS model.
**6 Conclusion**
The HIRESSS code (a physically-based distributed slope stability simulator for analysing shallow landslide triggering
conditions in real time and in large areas) was applied to the eastern sector of Valle d'Aosta region in order to test its
capability to forecast shallow landslides at regional scale. The model was applied in back analysis to two past rainfall
events that have triggered in the study areas several shallow landslides between 2008 and 2009. In order to run the
model and to increase its reliability, an in-depth study of the geotechnical and hydrological properties of hillslopes
controlling shallow landslides formation was conducted. In particular, two campaigns of on site measurements and
laboratory experiments were performed with 12 survey points. The data collected contributes to generate input map of
parameters for HIRESSS model according to lithological classes. The effect of vegetation on slope stability in terms of
root reinforcement has been also taken into account based on the plant species distribution and literature values of root
cohesion to product a map of root reinforcement of the study area. The outcomes of the model are daily failure
probability maps with a spatial resolution of 10 m. To evaluate the model performance both temporal and spatial
validation were carried out, and in general for both the simulated events the computed highest daily probability of
occurrence corresponds to the days and the areas of real landslides.

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

**Table 1.** Geotechnical properties of survey points (grain size distribution, Atterberg limits, index properties, permeability and shear strength parameters).

| SITE | SOIL TYPE | G % | S % | M % | C % | LL (%) | PL (%) | PI (%) | USCS | $\gamma$ (kN m⁻³) | $\gamma_d$ (kN m⁻³) | $\gamma_{sat}$ (kN m⁻³) | n (%) | w (%) | $k_s$ (m s⁻¹) | $k_{sc}$ (m s⁻¹) | $\phi'$ lab (°) | c' (kPa) |
|---|---|---|---|---|---|---|---|---|---|---|---|---|---|---|---|---|---|---|
| Site 1 | Sand with silty gravel | 27.8 | 45.2 | 23.4 | 3.6 | 36 | 25 | 11 | SM | 16.7 | 13.7 | 18.3 | 47.3 | 11.3 | / | 2.52E-06 | 25.6 | 1.0 |
| Site 2 | Sand with gravelly silt | 19.4 | 50.5 | 29.0 | 1.1 | 38 | 25 | 14 | SC | 19.1 | 14.5 | 18.8 | 44.3 | 11.4 | 2.71E-06 | 1.48E-06 | 34.3 | 1.5 |
| Site 3 | Sand with gravel and silt | 26.9 | 45.2 | 26.8 | 1.1 | / | / | / | / | / | / | / | / | / | / | / | 8.89E-07 | / | / |
| Site 4 | Sand with gravelly silt | 18.8 | 40.4 | 39.2 | 1.6 | 38 | 27 | 11 | SM | 19.5 | 14.8 | 19.0 | 43.2 | 10.7 | 1.36E-07 | 4.51E-07 | 34.3 | 0.0 |
| Site 5 | Sand with gravel and silt | 31.0 | 43.1 | 25.7 | 0.2 | 47 | 36 | 11 | SM | 18.4 | 14.0 | 18.5 | 46.3 | 11.0 | / | 2.44E-06 | 25.7 | 9.3 |
| Site 6 | Sand with poorly silty gravel | 28.5 | 57.5 | 13.9 | 0.1 | 52 | 38 | 13 | SM | 18.7 | 13.5 | 18.2 | 47.9 | 20.0 | / | 8.27E-06 | 30.2 | 4.4 |
| Site 7 | Sand with silty gravel | 37.0 | 42.6 | 17.9 | 2.5 | 40 | 32 | 8 | SM | 20.3 | 15.5 | 19.5 | 40.4 | 26.2 | 5.18E-06 | 2.97E-06 | 28.2 | 3.4 |
| Site 8 | Sandy silty gravel | 58.1 | 24.6 | 16.0 | 1.3 | 43 | 28 | 16 | GM | 17.2 | 15.7 | 19.6 | 39.6 | 9.4 | / | 3.76E-06 | 30.1 | 8.1 |
| Site 9 | Gravelly silty sand | 18.7 | 55.1 | 24.4 | 1.8 | 46 | 36 | 10 | SM | 20.1 | 18.7 | 21.5 | 27.9 | 8.1 | 2.41E-06 | 1.73E-06 | 33.9 | 0.6 |
| Site 10 | Sand with gravelly silt | 21.9 | 52.0 | 25.1 | 1 | 46 | 37 | 8 | SM | 18.4 | 16.0 | 19.8 | 38.6 | 15.5 | / | 2.10E-06 | 30.3 | 1.5 |
| Site 11 | Gravelly silty sand | 24.3 | 51.4 | 21.2 | 3.1 | 31 | 25 | 7 | SM | 21.7 | 18.0 | 21.2 | 31.9 | 20.5 | 4.03E-06 | 3.05E-06 | 29.8 | 2.0 |
| Site 12 | Gravel with poorly silty sand | 55.2 | 32.2 | 12.2 | 0.4 | 55 | 45 | 10 | SM | 15.3 | 14.6 | 18.9 | 43.9 | 5.1 | 1.54E-05 | 8.25E-06 | 30.2 | 1.6 |
| | MEAN | 30.63 | 44.98 | 22.9 | 1.48 | 42.91 | 32.18 | 10.82 | | 18.67 | 15.36 | 19.39 | 41.03 | 13.56 | 4.98E-06 | 3.16E-06 | 30.24 | 3.04 |
| | MEDIAN | 27.35 | 45.2 | 23.9 | 1.2 | 43 | 32 | 11 | | 18.7 | 14.8 | 19.0 | 43.2 | 11.3 | 3.37E-06 | 2.48E-06 | 30.2 | 1.6 |
| | STD.DEV | 13.31 | 9.48 | 7.41 | 1.11 | 7.15 | 6.71 | 2.71 | / | 1.80 | 1.68 | 1.10 | 6.34 | 6.30 | 5.38E-06 | 2.56E-06 | 3.05 | 3.07 |
| | MAX | 58.1 | 57.5 | 39.2 | 3.6 | 55 | 45 | 16 | | 21.7 | 18.7 | 21.5 | 47.9 | 26.2 | 1.54E-05 | 8.27E-06 | 34.3 | 9.3 |
| | MIN | 18.7 | 24.6 | 12.2 | 0.1 | 31 | 25 | 7 | | 15.3 | 13.5 | 18.2 | 27.9 | 5.1 | 1.36E-07 | 4.51E-07 | 25.6 | 0 |

**Table 2.** Spatialized geotechnical parameters of each lithological class as input for HIRESSS model.

| Lithological classes | Soil Type | $\phi'$ lab (°) | $c'$ (Pa) | $\gamma_d$ (kN m$^{-3}$) | $n$ (%) | $k_s$ (m s$^{-1}$) | $h_s$ | $q_r$ | $l$ |
|---|---|---|---|---|---|---|---|---|---|
| Calcareous schist | Sand with gravelly silt | 31 | 1000 | 16.5 | 39 | 1.1E-05 | 0.1466 | 0.041 | 0.322 |
| Alluvial deposits | Sand with gravel and silt | 26 | 1000 | 14.0 | 46 | 3.0E-06 | 0.1466 | 0.041 | 0.322 |
| Glacial deposits | Sand with silty gravel | 31 | 1000 | 15.3 | 41 | 2.7E-06 | 0.1466 | 0.041 | 0.322 |
| Colluvial deposits | Sand with silty gravel | 25 | 1000 | 13.7 | 47 | 2.5E-06 | 0.1466 | 0.041 | 0.322 |
| Granites | Sandy gravel | 30 | 1000 | 17.6 | 32 | 4.0E-06 | 0.1466 | 0.041 | 0.322 |
| Mica schists | Sandy silty gravel | 30 | 1000 | 17.7 | 32 | 6.0E-06 | 0.1466 | 0.041 | 0.322 |
| Green stones | Gravel with silty sand | 32 | 1000 | 16.3 | 37 | 4.6E-06 | 0.1466 | 0.041 | 0.322 |

**Table 3.** Main characteristics of the simulation.

|  | 2008 event | 2009 event |
|---|---|---|
| Spatial resolution | 10 m | 10 m |
| Time step | 1h | 1h |
| Rainfall hours | 192 | 96 |

5  **Table 4.** Hiresss results over 75% of slope failure probability for two events.

| Event 2008 | N. Pixel | Total % | Pixel area (km$^2$) |
|---|---|---|---|
| 24/05/2008 | 62344 | 1 | 6 |
| 25/05/2008 | 21295 | 0 | 2 |
| 26/05/2008 | 84256 | 1 | 8 |
| 27/05/2008 | 95220 | 1 | 10 |
| 28/05/2008 | 15364 | 0 | 2 |
| 29/05/2008 | 243137 | 3 | 24 |
| 30/05/2008 | 79437 | 1 | 8 |
| 31/05/2008 | 7110 | 0 | 1 |
| **Event 2009** | **N. Pixel** | **Total %** | **Pixel area (km$^2$)** |
| 25/04/2009 | 0 | 0 | 0 |
| 26/04/2009 | 52644 | 1 | 5 |
| 27/04/2009 | 326826 | 4 | 33 |
| 28/04/2009 | 56599 | 1 | 6 |

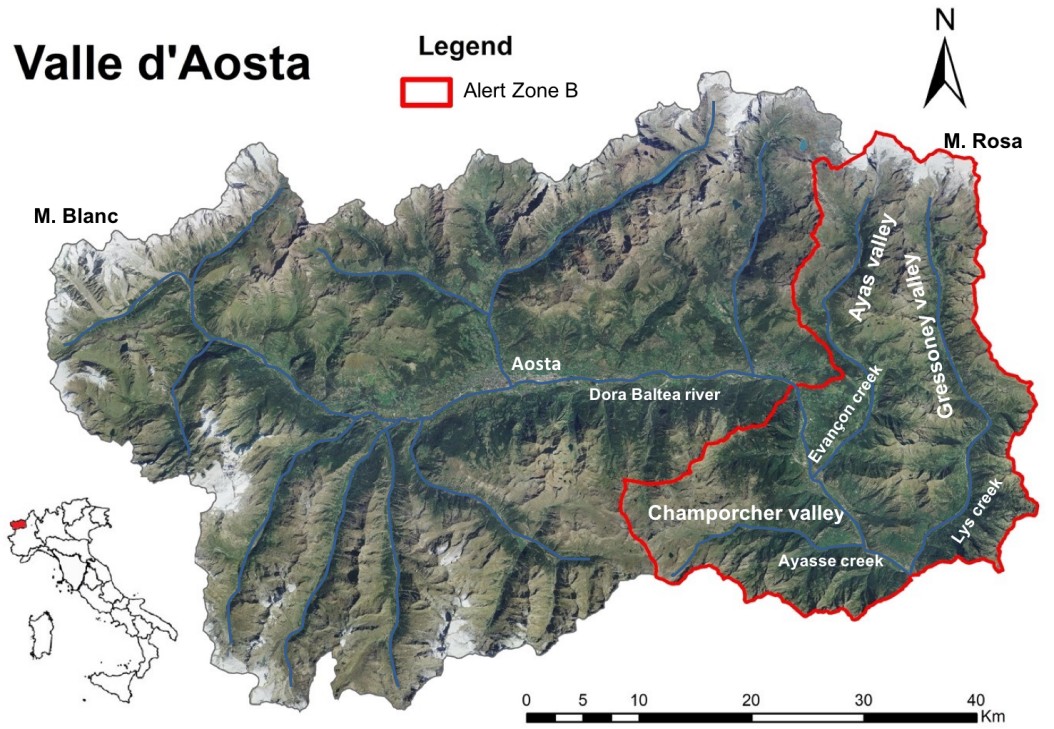

**Figure 1.** Valle d'Aosta region in the NW Italy: in red the study area, alert Zone B.

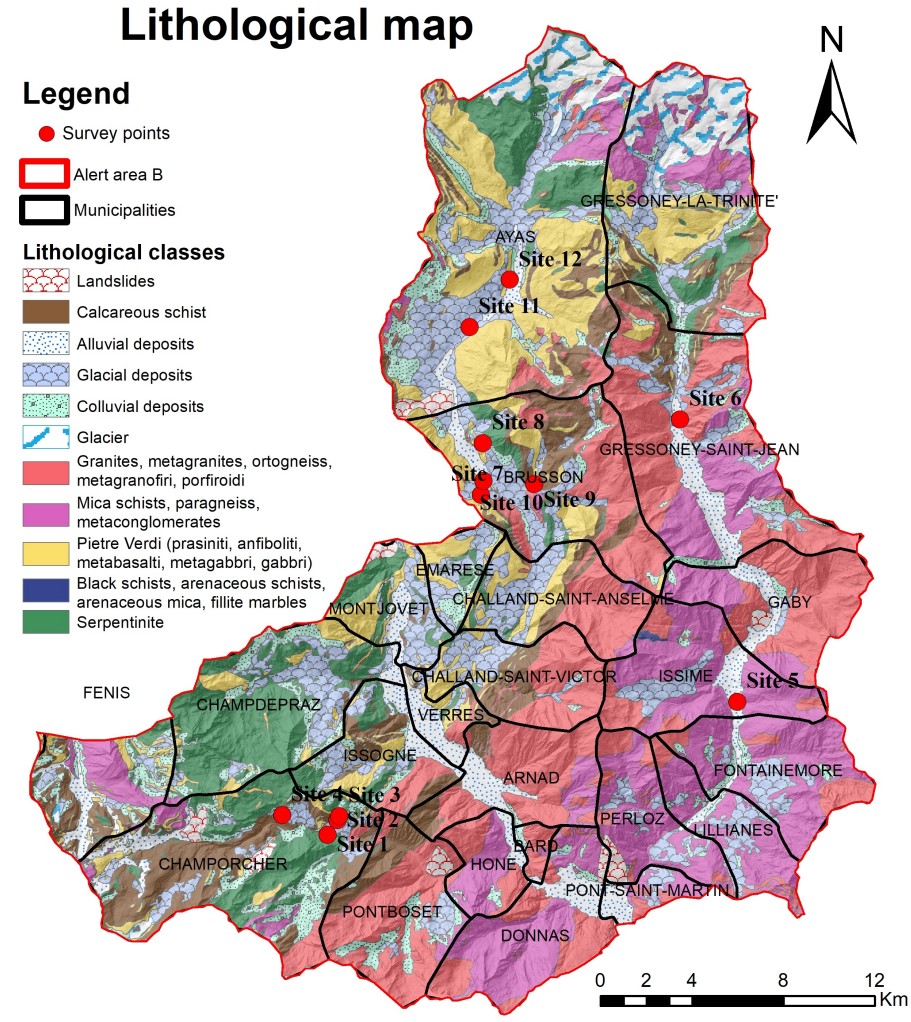

**Figure 2.** Spatial distribution of survey points compared to the geo-lithology.

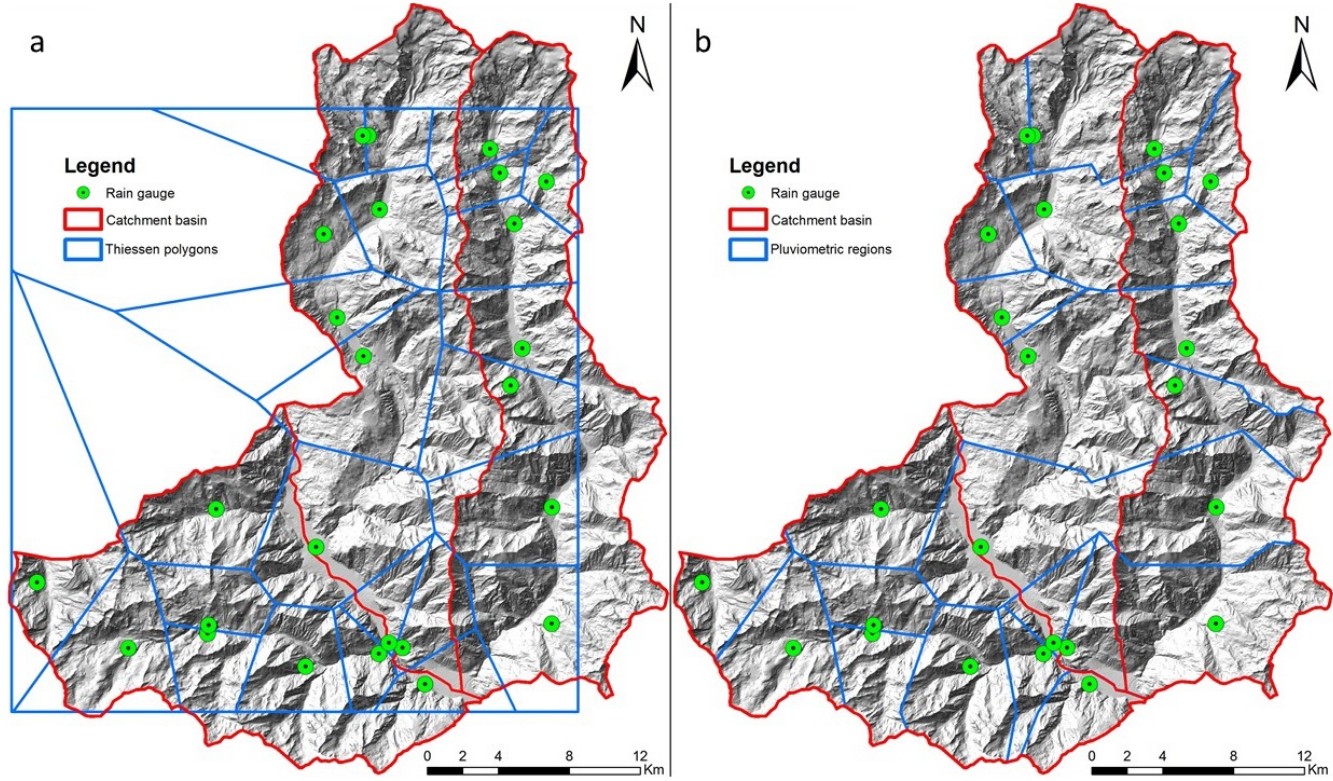

**Figure 3.** Comparison of Thiessen's polygons methodology a) simple b) modified according to the catchment basins boundaries.

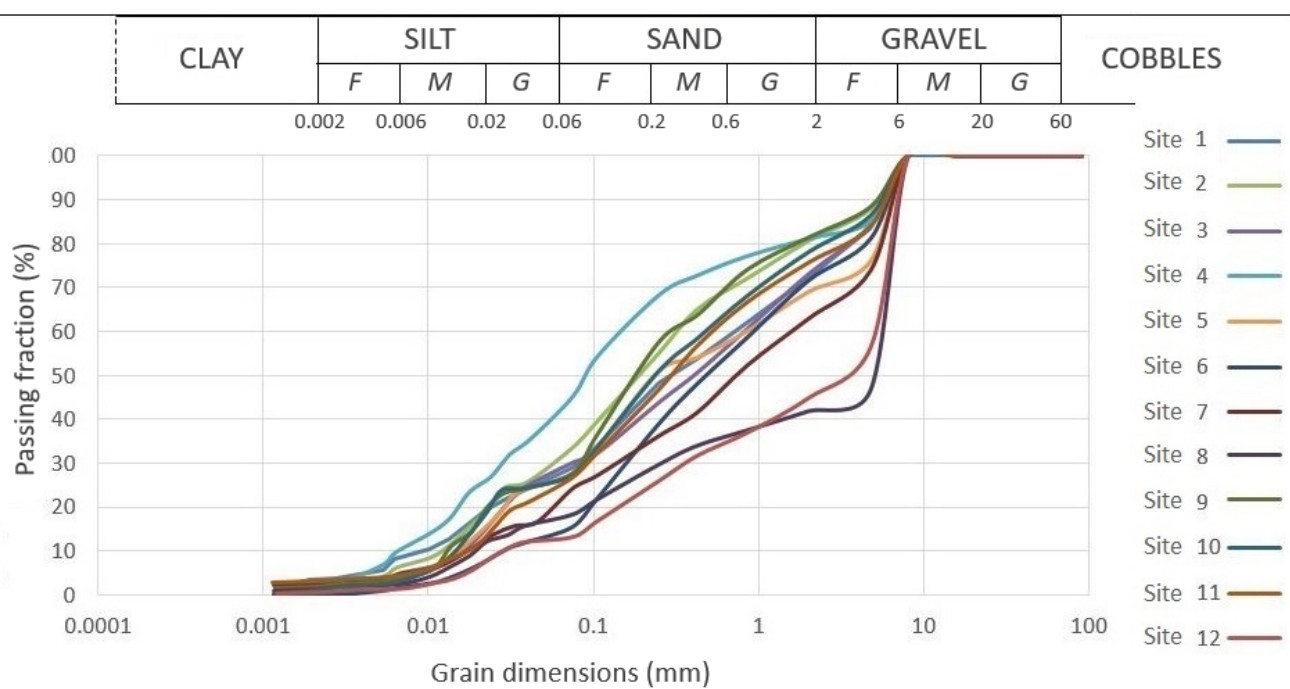

5    **Figure 4.** Grain size distributions of soil samples.

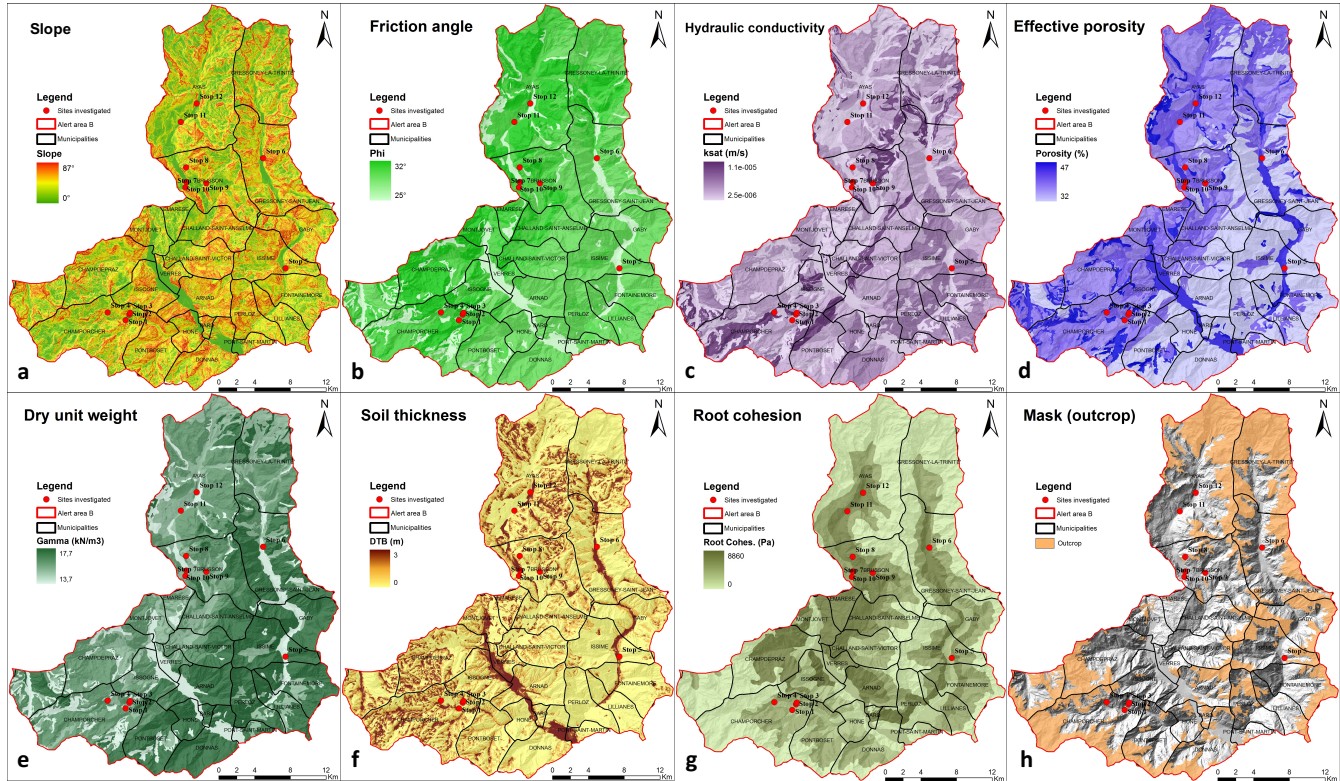

**Figure 5.** Static input parameters for HIRESSS model, a) slope gradient; b) friction angle;c) Hydraulic conductivity; d) effective porosity;e) dry unit weight; f) soil thickness; g) root cohesion; and h) exposure rock mask.

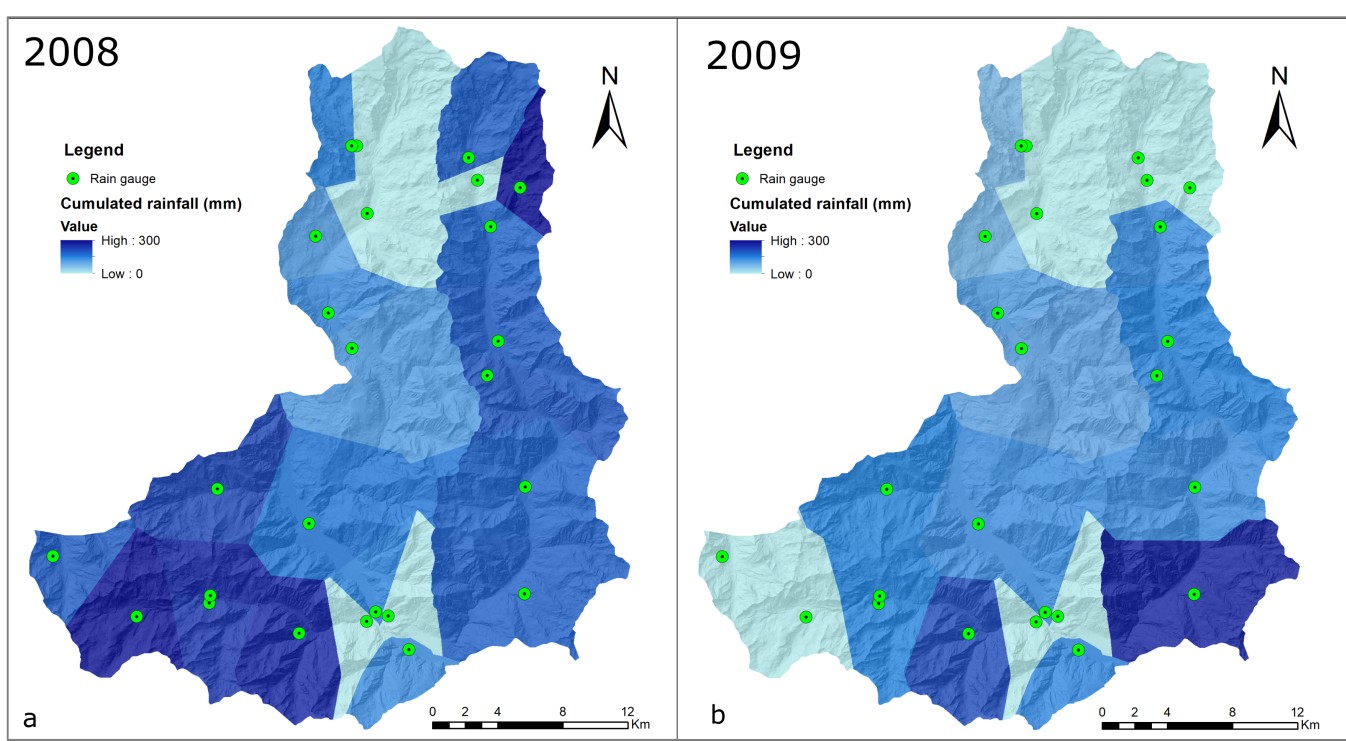

**Figure 6.** Cumulated rainfall maps for two events.

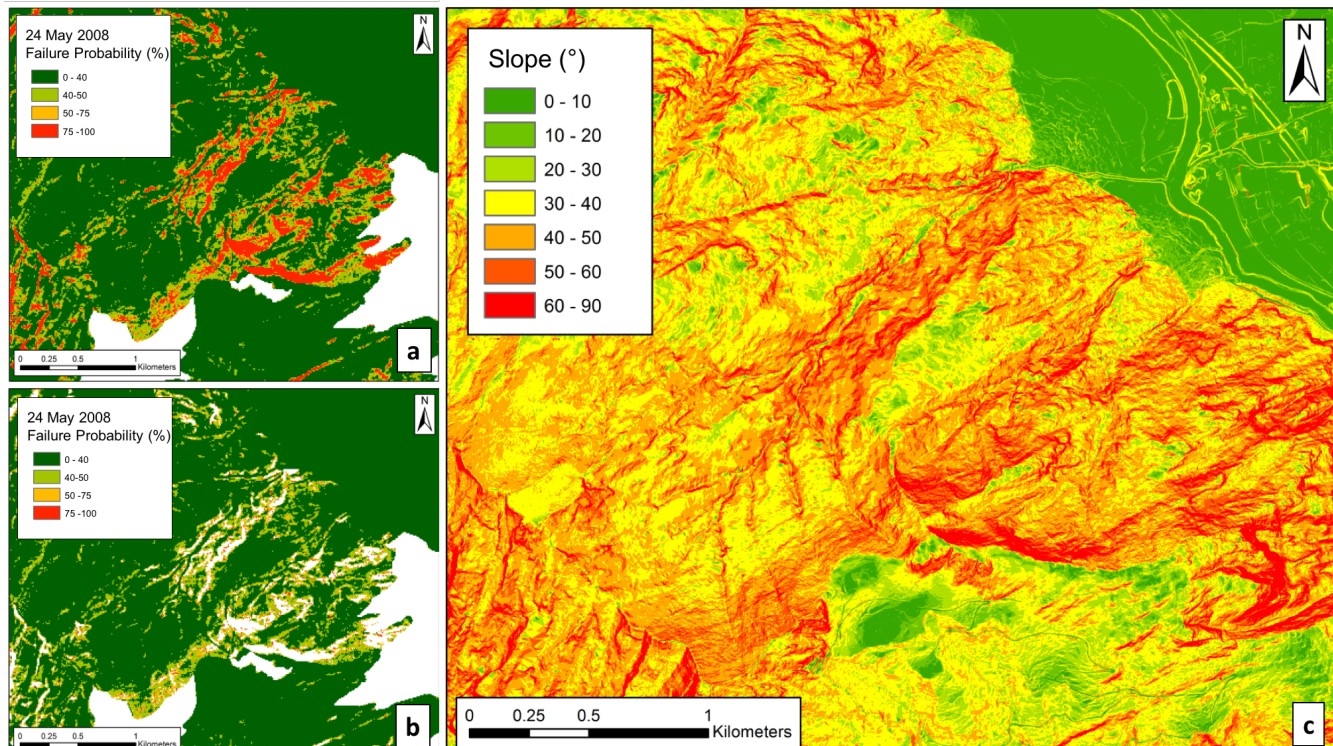

**Figure 7.** Example of numerical mask to remove the false positive of the first event simulated, between 24-31 May 2008, a) the HIRESSS result of the first day of simulation with false positive pixels, b) the probability map after the numerical mask implementation, c) the slope map shows that the pixels with high probability of landslide occurrence are located where the slope is higher than 60%.

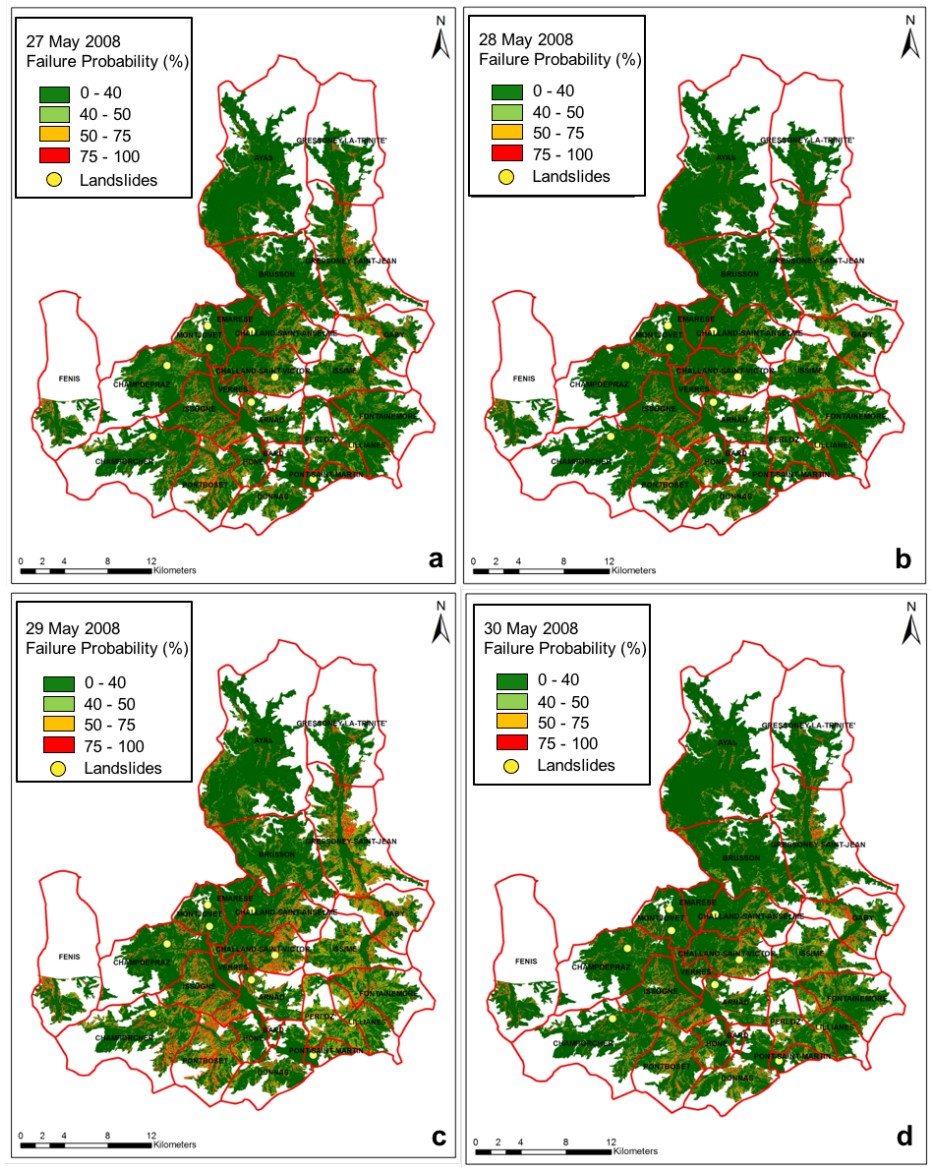

**Figure 8.** HIRESSS landslide probability maps of simulate event of 24-31 May 2008 and reporting landslide during this event focused on the four critical days, a) 27 May 2008, b) 28 May 2008, c) 29 May 2008, and d) 30 May 2008.

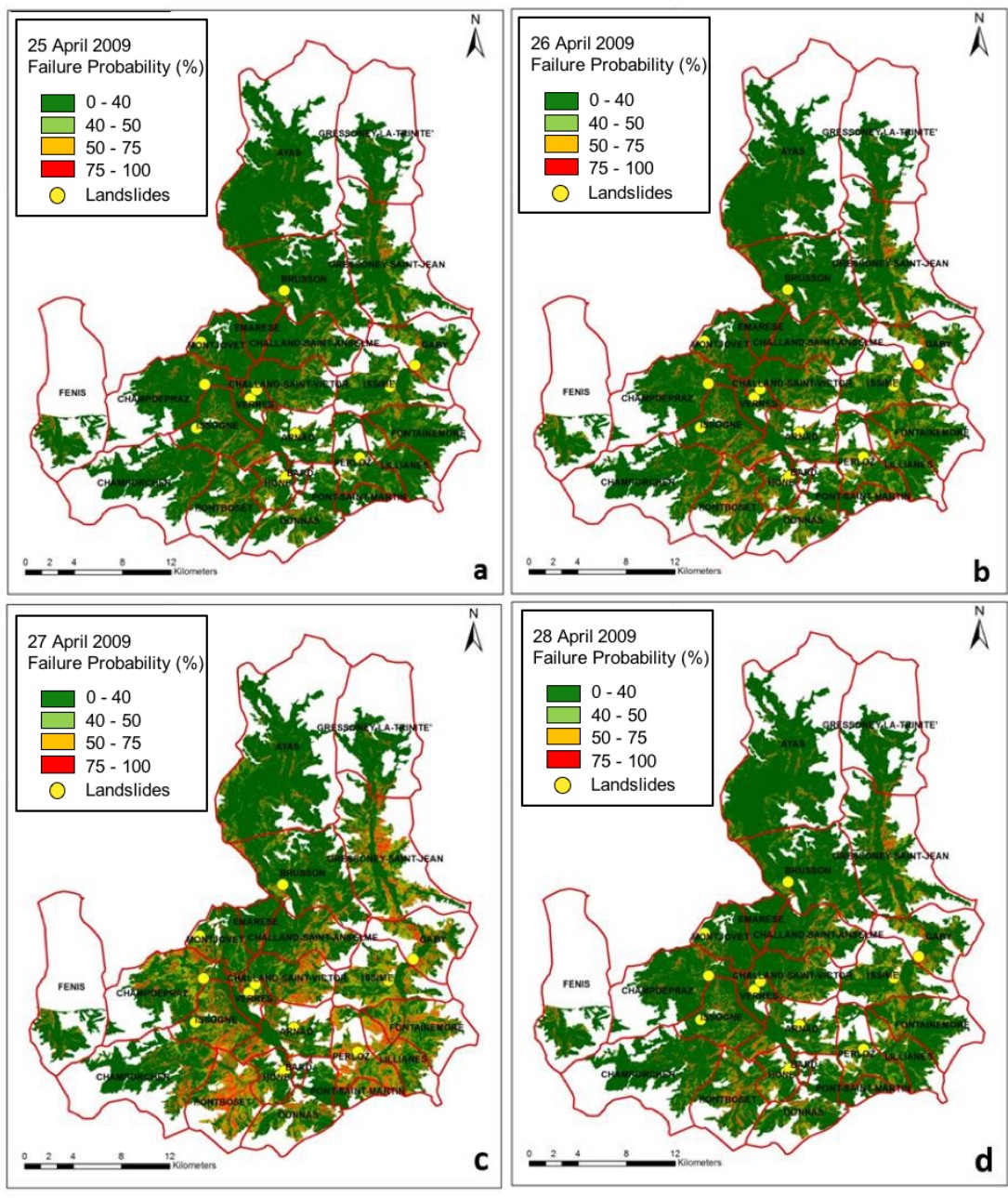

**Figure 9.** HIRESSS landslide probability maps of simulate event between 25 - 28 April 2009 and reporting landslide during this event, a) 25 April 2009, b) 26 April 2009, c) 27 April 2009 and d) 28 April, 2009.

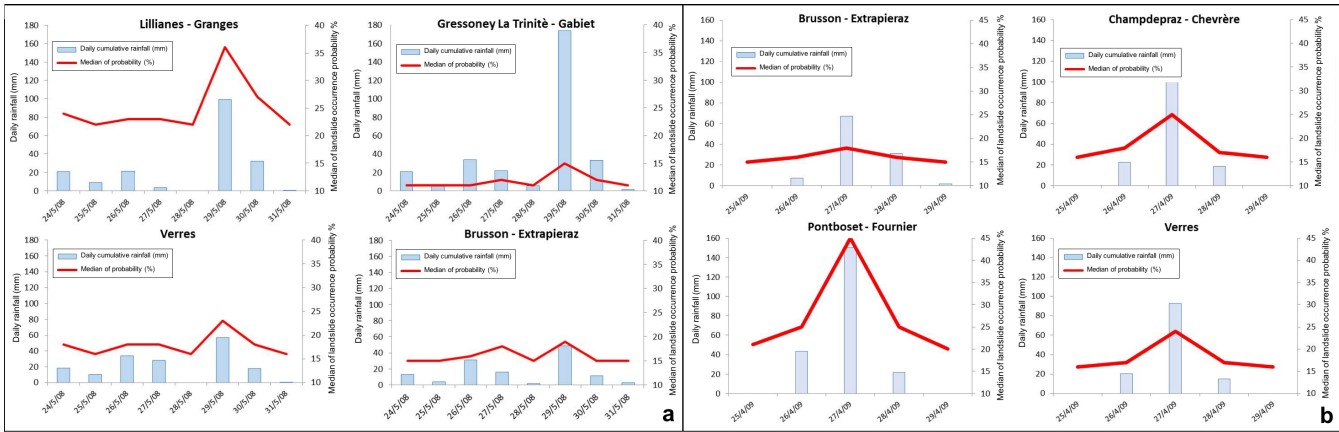

**Figure 10.** Correlation graphs between the daily cumulative rainfall and the median of landslide occurrence probability for both events.

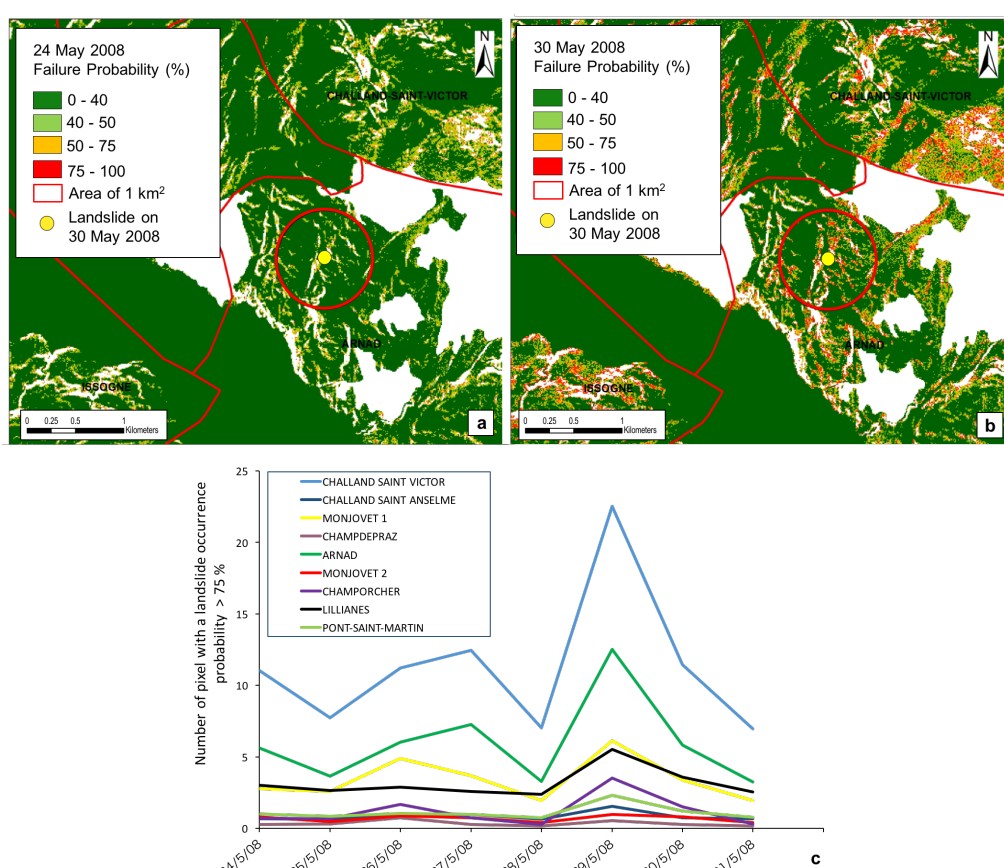

**Figure 11.** An example of landslide event happened in the Arnad municipality compared to landslide occurrence probability map, a) before and b) after rainfall event. c) Number of pixels above 75% of probability calculated by the model for all the landslides triggered during the event in the study area.

