# Peer review of "Application of a physically-based model to forecast shallow landslides occurrence at regional scale 2"

_Natural Hazards and Earth System Sciences, 2017_

## Referee Comment (RC1) · Anonymous Referee #1 · 11 Jan 2018

I think this manuscript presents the application of the HIRESS code to forecast shallow landslides at the regional scale. Especially the geotechnical and hydrological input data were measured in 12 sites and then the spatial distribution of measured data was estimated by Monte Carlo simulation. Through the application of HIRESS code, it is possible to forecast the shallow landslide using rainfall data in the special area with regional scale. So I think it deserves to be published in NHESS after some minor problems are solved clearly. Some minor problems are as follows;

Firstly, I wonder how to consider the unsaturated soil parameters such as bubbling pressure in the HIRESS code. I think the unsaturated soil parameters were not con-

[Figure]

sidered in this manuscript. As you know the shallow landslide is induced by the rain infiltration into the ground and saturation of the surface soil layer. To analyze this phenomenon, the relationship between matric suction and water contents in the surface soil layer was considered in a view of unsaturated soil mechanism.

Second, to make Thiessen's polygons for the rainfall data in a certain area, the rainfall data in study area as well as out of the study area especially around the study area should be used. But, in this study, the rainfall data in the only study area were used to make Thiessen's polygons. Also, the modification method of Thiessen's polygons should be verified.

Finally, in this manuscript, the final aim is to set-up the early warning system for shallow landslide with regional scale. But this manuscript focused on the application of the HIRESS code to the special area to forecast shallow landslide. Therefore, this part should be corrected and complemented to match up with the overall contents of the manuscript.

---

## Referee Comment (RC2) · Anonymous Referee #2 · 28 Jan 2018

The topic of the work meets the scope of the journal well. However, it is difficult for readers to recognize its contributions to the science community from its title, abstract and even the introduction part. The Introduction, Methodology and Discussion sections are not well-structured and pose difficult for readers to understand. My specific concerns are listed below:

1) The introduction part fails to convey the current research gap and readers have difficulty to assess its scientific significance. It is unable to convince the readers why the authors carry out this work. It seems that authors want to share with the community some improvements by considering soil and vegetation parameters by using the exist-

[Figure]

ing model HIRESSS. I recommend the authors first detail the research question clearly, and then briefly describe their way to solve the problem.

2) The Methodology part is mixed with Results. For example, lines 123-135 were measured results.

3) The structure of the Methodology is not logical. I suggest the authors put an outline paragraph at the beginning of this section, in which they brief the logics of this section. "3.3 HIRESSS description" and "3.4 HIRESSS input data" should be placed in the beginning of the Methodology.

4) Although physically based landslide model is desirable, the input data is enormous and rigorous. The data of root cohesion and some of the soil values seem to be derived from existing literature review. Is it really proper to directly use these data in your study? You should justify this problem.

5) Please detail the acquired time, spatial resolution and other characteristics of the DEM used in the model.

6) The discussion part is poorly written. Authors should explain the results, compare with other's work, provide implications, acknowledge its limitations and echo the introduction part. I think this part should be significantly improved.

---

## Referee Comment (RC3) · Anonymous Referee #3 · 4 Feb 2018

The paper corresponds to the journal scope. In a general point of view: the paper is not very well structured, it is difficult for the reader to understand the message of the authors and to follow the text. The text lacks of consistency and some improvements are requested in order to publish the paper. Some recent references has to include and some sentences should be simplified. More precisely and point by point: 1. The abstract and the introduction have to be rewritten. For instance, the problematic is not visible. The authors have to put the problem(s), the solution in general (with a state of the art) and after the contribution of their research. Clarify the introduction please. 2. The geographical description has to be modified. The description is not straightforward. In general you can start by the geological context with the lithology and the structure

and after the landscape and the geomorphology of the area. After you follow by the weather and if you have information by landuse. 3. The methodology is not very well described, please revised it with a part about the HIRESS model, and after HIRESS data. The problem of root reinforcement can be put in the introduction or if you want absolutely speak about this topic, make a part called " background". Moreover, the part about data is few explained. Improve it please. 4. I think there is some lack of description about the root influence in your model and the way to obtain these information. 5. I think the monte carlo approach coupled with uncertainty is not new for landslide susceptibility assessment with PBM, there are some references to include in your text as Mergili et al., 2014 or Thiery et al., 2017 with r. slope. stability or ALICE tool used this approach to integrate the uncertainty of environmeent (geotechnical values). You have to mention these references in your text. doi:10.1016/j.geomorph.2013.10.008 or Thiery et al. : Thiery, Y., Vandromme, R., Maquaire, O., Berneradie, S., 2017. HYPERLINK http://link.springer.com/chapter/10.1007/978-3-319-53498-5_104" Landslide susceptibility assessment by EPBM (Expert physically based model): strategy of calibration in complex environment. In: Mikoš, M., Tiwari, B., Yin, Y., Sassa, K. (Eds) Advancing Culture of Living with Landslides. Proceedings, Vol. 2: Advances in Landslide Science, Springer, 4th World Landslide Forum in Ljubljana, pp.917-926. https://doi.org/10.1007/978-3-319-53498-5_104. You can mention the last paper with TRIGRS :https://doi.org/10.1007/s10346-017-0931-7 6. Finally, the discussion is not a discussion. In a scientific paper the discussion emphasize the results, the advantages of the method but also the drawbacks, the comparison with another approaches. In your text, there are any comments like that.

We suggest another structure of the text as follow: 1. Study sites 2. Background (if you choose this way) 3. Model: description, improvement and the strategy used (calibration, etc) 4. Data used or created for your study 5. Results 6. Discussion 7. Conclusion

Please also note the supplement to this comment:
https://www.nat-hazards-earth-syst-sci-discuss.net/nhess-2017-425/nhess-2017-425-

RC3-supplement.zip

---

## Referee Comment (RC4) · Anonymous Referee #3 · 4 Feb 2018

One point not discussed in my first comments: Where is the relation with early warning system?It is a little explianed but it is not justify the term "early warning" in the tittle. I think you have to improve this topic in your text if you want to improve your it.

---

## Author Response (AR1)

*I think this manuscript presents the application of the HIRESS code to forecast shallow landslides at the regional scale. Especially the geotechnical and hydrological input data were measured in 12 sites and then the spatial distribution of measured data was estimated by Monte Carlo simulation. Through the application of HIRESS code, it is possible to forecast the shallow landslide using rainfall data in the special area with regional scale. So I think it deserves to be published in NHESS after some minor problems are solved clearly. Some minor problems are as follows;*

AC: We thank the referee with his/her revision and fruitful comments.

**1:** *Firstly, I wonder how to consider the unsaturated soil parameters such as bubbling pressure in the HIRESS code. I think the unsaturated soil parameters were not considered in this manuscript. As you know the shallow landslide is induced by the rain infiltration into the ground and saturation of the surface soil layer. To analyze this phenomenon, the relationship between matric suction and water contents in the surface soil layer was considered in a view of unsaturated soil mechanism.*

AC: We thank the referee for the comment but we are not sure to have properly understood the comment. In particular, we are not sure if the comment wants to highlight that the unsaturated parameters were not considered in the analysis. If this is the key point we want to stress that the HIRESSS model considers the effect of matric suction in unsaturated soils, taking into account the increase in strength and cohesion. The variation of matric suction based on volumetric water content, defined trough the hydrological model, is modelled taking into account the parameters of the soil characteristic curves (the bubbling pressure, the pore size index distribution and the residual water content). Unfortunately, we have not defined the soil characteristic curve experimentally but the soil characteristic curves parameters were derived from literature values (Rawls et al., 1982) based on the soil types measured through laboratory analysis. We will revise the text providing a more clear and in-depth explanation on how the parameters of unsaturated soils have been taken into account in the analysis.

**2:** *Second, to make Thiessen's polygons for the rainfall data in a certain area, the rainfall data in study area as well as out of the study area especially around the study area should be used. But, in this study, the rainfall data in the only study area were used to make Thiessen's polygons. Also, the modification method of Thiessen's polygons should be verified.*

AC: To properly run the HIRESSS model we needed spatially distributed rainfall data; the most obvious approach could be the use of a geostatistical model to interpolate rainfall data (e.g. IDW or Kriging), but these approaches are not suitable for the study area, because of the morphology of the territory (small valleys surrounded by high mountains), that is not considered in these models. So, we deiced to define a sort of "relevance area" of each rain gauge and the same rainfall value (for each hourly time step) has been assigned inside each area.

We used only rain gauges of the study area because we did not have other stations to be used in the definition of the Thiessen's polygons.

The modification of polygons has been carried out to take into account the morphology of the area and to avoid that data of some rain gauges could be considered in different river basins.

**3:** *Finally, in this manuscript, the final aim is to set-up the early warning system for shallow landslide with regional scale. But this manuscript focused on the application of the HIRESS code to the special area to forecast shallow landslide. Therefore, this part should be corrected and complemented to match up with the overall contents of the manuscript.*

AC: As discussed also in the introduction of the manuscript, warning systems for landslides can be designed and employed at different reference scales. In particular local systems for single slopes and regional systems. Usually the term regional refers to an area bigger than the single slope. Here below a list of selected references that report regional application of physically based models:

Baum, R., Savage, W., Godt, J., 2002. Trigrs: A FORTRAN program for transient rainfall infiltration and grid-based regional slope- stability 322 analysis, Open-file Report, US Geol. Survey.

Baum, R.L., Godt, J.W., Savage, W.Z., 2010. Estimating the timing and location of shallow rainfall-induced landslides using a model for transient unsaturated infiltration. J Geophys Res 115:F03013.

Chen, H.X., Zhang, L.M., 2014. A physically-based distributed cell model for predicting regional rainfall-induced shallow slope failures. Engineering Geology doi:10.1016/j.enggeo.2014.04.011

Dietrich, W., Montgomery, D., 1998. Shalstab: a digital terrain model for mapping shallow landslide potential. NCASI (National Council for Air and Stream Improvement) Technical Report, February, 1998.

Rossi, G., Catani, F., Leoni, L., Segoni, S., Tofani, V., 2013. HIRESSS: a physically based slope stability simulator for HPC applications. Nat. Hazards Earth Syst. Sci., 13, pp. 151–166.

Salciarini, D., Fanelli, G., Tamagnini, C., 2017. A probabilistic model for rainfall-induced shallow landslide prediction at the regional scale, 386 Landslides, 14(5),1731–1746.

We think that the term regional is appropriate and it can be left in the manuscript.

**Anonymous Referee #2

*The topic of the work meets the scope of the journal well. However, it is difficult for readers to recognize its contributions to the science community from its title, abstract and even the introduction part. The Introduction, Methodology and Discussion sections are not well-structured and pose difficult for readers to understand.*

AC: We would like to thank the referee for his/her careful revision and fruitful comments. We agree with the referee that the manuscript needs an in-depth revision especially concerning the structure and organization of the sessions. We are currently working in this direction and we are completely reorganization the contents of the Introduction, methodology and discussion.

*My specific concerns are listed below:*

**1)** *The introduction part fails to convey the current research gap and readers have dif- ficulty to assess its scientific significance. It is unable to convince the readers why the authors carry out this work. It seems that authors want to share with the community some improvements by considering soil and vegetation parameters by using the existing model HIRESSS. I recommend the authors first detail the research question clearly, and then briefly describe their way to solve the problem*

AC: We thank he referee for the comments. We are rewriting the Introduction, trying to highlight better our key research questions and which are the main objectives of the research work. Our main objective is to test the application of an, already developed, physically-based model to forecast the occurrence of shallow landslides in a selected case study. Furthermore the work wants to highlight some improvements related to the soil parameters characterization and contribution of vegetation to slope stability. In order to be consistent between title and contents of the manuscript we propose to change the title
from: Regional physically based landslide early warning modelling: soil parameterisation and validation of the results.
to:  Application of a physically-based model to forecast shallow landslides occurrence at regional scale.

**2)** *The Methodology part is mixed with Results. For example, lines 123-135 were measured results.*

AC: We agree with the referee and we are currently restructuring the text in order to separate methodology and results.

**3)** *The structure of the Methodology is not logical. I suggest the authors put an outline paragraph at the beginning of this section, in which they brief the logics of this section. "3.3 HIRESSS description" and "3.4 HIRESSS input data" should be placed in the beginning of the Methodology.*

AC: Again we agree with the referee. The methodological part has being revised in order to be more readable and clear.

**4)** *Although physically based landslide model is desirable, the input data is enormous and rigorous. The data of root cohesion and some of the soil values seem to be derived from existing literature review. Is it really proper to directly use these data in your study? You should justify this problem.*
AC: The physically based models require many hydrological and geotechnical parameters as input data. In many cases, for each geotechnical parameter, a constant value is used for the whole study area as averaged from in situ measurements or derived from literature data. In some studies, a limited degree of spatial variability is ensured using a certain value for distinct geological, lithological, or engineering geological units, as derived from direct measurements or from existing databases and published data.

In this work we have tried to characterize as much as possible the soil covers from a hydrological and geotechnical point of view, through several direct in-situ and laboratory measurements. In particular the measured parameters are: effective cohesion, friction angle, dry unit weight, hydraulic conductivity effective porosity.

Some other parameters have not been measured, in particular we have not defined the soil characteristic curve experimentally but the soil characteristic curves parameters were derived from literature values based on the soil types measured through laboratory analysis.

At the same time the experimental evaluation of root cohesion is quite complicated and time demanding and we have chosen to define this value based on relevant literature for the different types of vegetation cover.

We will explain better this issue in the text and we will critically examine it in the discussion.

**5)** *Please detail the acquired time, spatial resolution and other characteristics of the DEM used in the model.*

AC: We will add this information in the text.

**6)** *The discussion part is poorly written. Authors should explain the results, compare with other's work, provide implications, acknowledge its limitations and echo the intro- duction part. I think this part should be significantly improved.*

AC: As already said before we are completely reorganizing the discussion session.

**Anonymous Referee #3

*The paper corresponds to the journal scope. In a general point of view: the paper is not very well structured, it is difficult for the reader to understand the message of the authors and to follow the text. The text lacks of consistency and some improvements are requested in order to publish the paper. Some recent references has to include and some sentences should be simplified. More precisely and point by point:*

AC: Dear Referee, Thanks for your detailed revision. We agree that the manuscript needs a general reorganization of the structure, with special reference to the methodology and discussion of the results. We are currently working in this direction and we are completely reorganization the contents of the introduction, methodology and discussion.

**1.** *The abstract and the introduction have to be rewritten. For instance, the problematic is not visible. The authors have to put the problem(s), the solution in general (with a state of the art) and after the contribution of their research. Clarify the introduction please.*

AC: We thank he referee for the comment. We are rewriting the Introduction, trying to highlight better our key research questions and which are the main objectives of the research work.
Our aim is to test the application of an already developed, physically based model to forecast the occurrence of shallow landslides in a selected case study in Italy. Furthermore the work wants to highlight some model improvements related to the soil parameters characterization and contribution of vegetation to slope stability

**2.** *The geographical description has to be modified. The description is not straightforward. In general you can start by the geological context with the lithology and the structure and after the landscape and the geomorphology of the area. After you follow by the weather and if you have information by land use.*

AC: We agree; we are modifying this part according to the referee comment.

**3.** *The methodology is not very well described, please revised it with a part about the HIRESS model, and after HIRESS data. The problem of root reinforcement can be put in the introduction or if you want absolutely speak about this topic, make a part called " background". Moreover, the part about data is few explained. Improve it please.*

AC: We agree, we are revisiting this part. The methodology will start with the description of the HIRESSS model and then the input data. The problem of root reinforcement will be treated in the Introduction and then we will describe in the methodology how we have taken into account this parameter in our model.

**4.** *I think there is some lack of description about the root influence in your model and the way to obtain these information.*

AC: The problem of root reinforcement will be treated in the Introduction and then we will describe in the methodology how we have taken into account this parameter in our model.

**5.** *I think the monte carlo approach coupled with uncertainty is not new for landslide susceptibility assessment with PBM, there are some references to include in your text as Mergili et al., 2014 or Thiery et al., 2017 with r. slope. stability or ALICE tool used this approach to integrate the uncertainty of environmeent (geotechnical values). You have to mention these references in your text. doi:10.1016/j.geomorph.2013.10.008 or Thiery et al. : Thiery, Y., Vandromme, R., Maquaire, O., Berneradie, S., 2017. HYPERLINK http://link.springer.com/chapter/10.1007/978-3-319-53498-5_104"Land- slide susceptibility assessment by EPBM (Expert physically based model): strategy of calibration in complex environment. In: Mikoš, M., Tiwari, B., Yin, Y., Sassa, K. (Eds) Advancing Culture of Living with Landslides. Proceedings, Vol. 2: Advances in Landslide Science, Springer, 4th World Landslide Forum in Ljubljana, pp.917-926. https://doi.org/10.1007/978-3-319-53498-5_104. You can mention the last paper with TRIGRS :https://doi.org/10.1007/s10346-017-0931-7 6.*

AC: We thank the referee and we will include these references in the text.

**6:** *Finally, the discussion is not a discussion. In a scientific paper the discussion emphasize the results, the advantages of the method but also the drawbacks, the comparison with another approaches. In your text, there are any comments like that.*
*We suggest another structure of the text as follow: 1. Study sites 2. Background (if you choose this way) 3. Model: description, improvement and the strategy used (calibra- tion, etc) 4. Data used or created for your study 5. Results 6. Discussion 7. Conclusion.*
*One point not discussed in my first comments: Where is the relation with early warning system?It is a little explianed but it is not justify the term "early warning" in the tittle. I think you have to improve this topic in your text if you want to improve your it.*

AC: We thank the referee for the fruitful comment. We are completely reorganizing the text and consequently the structure of the manuscript.

**7:** *One point not discussed in my first comments: Where is the relation with early warning system?It is a little explianed but it is not justify the term "early warning" in the tittle. I think you have to improve this topic in your text if you want to improve your it.*

AC: In order to be consistent between title and contents of the manuscript we propose to change the title
From: Regional physically based landslide early warning modelling: soil parameterisation and validation of the results.
To: Application of a physically-based model to forecast shallow landslides occurrence at regional scale.

**Résumé des commentaires sur nhess-2017-425_reviewer03.pdf**

**Page : 1**

*Auteur : Sujet : Texte surligné Date : 28/01/2018 17:06:43 please, simplify the text. one sentence is sufficient*

AC: Done

*Auteur : Sujet : Texte surligné Date : 28/01/2018 17:11:09*

*I think you can start the sentence by another term like: it is possible to define reliable alert levels by statistical analysis of failure probability.*

AC: This part has been removed.

**Page : 2**

*Auteur : Sujet : Texte surligné Date : 28/01/2018 17:13:45 simplify the text please, the sentence is hard to understand in one read*

AC: Done

*Auteur : Sujet : Texte surligné Date : 04/02/2018 18:01:18*

AC: The introduction is now change in many parts as required.

*Auteur : Sujet : Texte surligné Date : 28/01/2018 17:14:41 for shallow landslide ? for deep landslide? please re-precise*

AC: The introduction is now change in many parts as required.

*Auteur : Sujet : Texte surligné Date : 28/01/2018 17:16:18*

*I think you can simplify the tex by one sentence. You hav etoo barrative sentence, please go to the subject straightaway*

AC: The introduction is now change in many parts as required.

*Auteur : Sujet : Texte surligné Date : 28/01/2018 17:18:38*

*Which context do you speak ? vegetation mitigation is not always used by geotechnical office. in lot of cases the solution are not based on vegetation solutions reinforcement, wall, etc....). please revise this part of the text.*

AC: The introduction is now change in many parts as required.

*Auteur : Sujet : Texte surligné Date : 28/01/2018 17:20:19*

*ok, but it is a specific case. you srudy site is in this context ? if not maybe you can improve your introduction by giving the different context of vegetation solution and related context they are used.*

*AC: The introduction is now change in many parts as required.*

*Auteur : Sujet : Texte surligné Date : 28/01/2018 17:22:40*

*ok, but please put transition sentence. waht is the relation between the text about root reinforcement and he used of HIRESS ? where is the problematic in this introduction ?*

AC: The introduction is now change in many parts as required.

**Page : 3**

*Auteur : Sujet : Note Date : 28/01/2018 17:23:54*

*there is a lack of one problematic in the introduction. Please revise it.*

AC: Tanks as we explained in the general comments, we proceeded to change the introduction in many parts, inserting the objectives of the work.

*Auteur : Sujet : Texte surligné Date : 28/01/2018 17:25:50*

*ok, but what do you mean by high climatic variations? rainfall? others? please give some information*

AC: The climate of the region is characterized by high variability strongly influenced by altitude (ranging from 400 m a.s.l of Dora Baltea's river floodplain to 4810 m a.s.l. of Mont Blanc), with a continental climate in the valleys floor and an Alpin climate at high altitudes.

*Auteur : Sujet : Note Date : 28/01/2018 17:29:51*

*In a geographical description, the best is to start with geological settings which give the structure of the landscape, after the geoporphological processes are given in order to explain the landscape formation since the last glaciation. please revise your text.*

AC: Thank you for the comments we modified the text as required: "From a geological point of view, the Valle d'Aosta is located NW with respect to the Insubrica Line, in particular, there are three systems of Europa chain: the Austroalpino, the Pennidiche and the Elvetico-Ultraelevato systems (De Giusti, 2004). Fig. 2 shows the lithological map of the study area obtained by reclassifying the geological units according to 11 lithological groups: landslides, calcareous schist, alluvial deposits, glacial deposits, colluvial deposits, glacier, granites, mica schists, green stone, black schists and serpentinites. In detail in the study area the main lithologies outcropping are metamorphic and intrusive rocks, in particular granites, metagranites, schists and serpentinite.

The geomorphology of the region is characterized by steep slopes and valleys shaped by glaciers. The glacial modelling is shown in the U-shaped of Lys and Ayas valleys, and the erosive depositional forms found in the Ayas valley. The three valleys' watercourses, the Lys creek, the Evançon creek, and the Dora Baltea river, contributed to the glacial deposits modelling with the formation of alluvial fans. The climate of the region is characterized by high variability strongly influenced by altitude (ranging from 400 m a.s.l of Dora Baltea's river floodplain to 4810 m a.s.l. of Mont Blanc), with a continental climate in the valleys floor and an Alpin climate at high altitudes."

*Auteur : Sujet : Texte surligné Date : 28/01/2018 17:32:17*

*for landslides have you some references ? how do you know the processes, their typology etc... moreover more explanations about shallow landslides (this is the object of the early warning system you propose) shlould be welcome. please detail landslides phenomenon.*

AC: The informations about landslide have been taken from Catasto dei Dissesti Regionale –Val d'Aosta, http://catastodissesti.partout.it, we added some information about the shallow landslide considered: "The slope steepness, together with mean annual precipitation of 800-900 mm are the main landslide triggering factors. These features lead the study area to be prone to landsliding, in particular rock falls, deep seated gravitational slope deformations (DSGSD), rocks avalanches, debris avalanches, debris flows, and debris slides (Catasto dei Dissesti Regionale – form Val d'Aosta Regional Authorities). In this work we model the triggering conditions of shallow landslides, i.e. soil slips and translational slides and we do not take into account the other types of movement."

**Page : 4**

*Auteur : Sujet : Note Date : 28/01/2018 17:36:37*

*I don't understand how you have structured the text. I think you have to give the detail of each formations before, which formation have been investigated?*

AC: We change the structure of the text as suggested, the methodology will start with the description of the HIRESSS model and then the input data preparation.

*Auteur : Sujet : Texte surligné Date : 28/01/2018 17:40:17*

*You chose the slope deposits point to investigate by analysis of DTM?*

*I am very surprised, for me the field survey, observations is the first way to chose good locations.*

*Have you make a dteail geomorphological analysis of the study sites? This is the first step to conduct a slope instability analysis.*

AC: We wanted know some field informatinos about the properties of soil deposits and so we chose some survey points based on geographic, lithological information and on landslide map. When we were on field some accessible areas were private and therefore it was not possible to analyze them.

*Auteur : Sujet : Texte surligné Date : 28/01/2018 17:42:05*

*you can simplify. Please make one reference about the protocol. if the reader wants more information, he can read the protocol on another paper.*

AC: Done, we simplify the text and insert some protocol reference.

**Page : 5**

*Auteur : Sujet : Texte surligné Date : 28/01/2018 17:45:12*

*you have to reduce this part, lot of thing can be read in litterature. Simplify the text.*

AC: Done thank you for the comment The root reinforcement is insert as static data in the section 3.2 HIRESSS Input data preparation.

**Page : 6**

*Auteur : Sujet : Texte surligné Date : 28/01/2018 17:47:36*

*simplify the text about roots etc ...*

*AC:* Done

**Page : 7**

*Auteur : Sujet : Texte surligné Date : 04/02/2018 18:00:48*

*the approach is not new, Mergili et al., 2014 or Thiery et al., 2017 with r. slope. stability or ALICE tool used this approach to integrate the uncertainty of environmeent (geotechnical values). You have to mention these references in your text.*

*doi:10.1016/j.geomorph.2013.10.008*

*Thiery et al.*

*Thiery, Y., Vandromme, R., Maquaire, O., Berneradie, S., 2017. HYPERLINK "http://link.springer.com/ chapter/10.1007/978-3-319-53498-5_104" Landslide susceptibility assessment by EPBM (Expert physically based model): strategy of calibration in complex environment. In: Mikoš, M., Tiwari, B., Yin, Y., Sassa, K. (Eds) Advancing Culture of Living with Landslides. Proceedings, Vol. 2: Advances in Landslide Science, Springer, 4th World Landslide Forum in Ljubljana, pp.917-926. https://doi. org/10.1007/978-3-319-53498-5_104*

*you can mention the last paper with TRIGRS :https://doi.org/10.1007/s10346-017-0931-7*

AC: The use of Monte Carlo Simulation inside the HIRESSS code is not new, it is just explained in the work of Rossi et al.(2013), we also include the suggested reference in the text.

**Page : 8**

*Auteur : Sujet : Note Date : 28/01/2018 18:01:15*

*I think you have a problem of structure in your text. I think it is bettre to present 1. study sites*

*2 Model, iprovement and the strategy used (calibatrtion etc)*

*3 data used or created for your study*

*4 results*

*The text will be more clear and understandable*

AC: Thank you for the comment, this part is also recommended by the other rewires we have providds to change the structure of the paper as suggested.

*Auteur : Sujet : Texte surligné Date : 28/01/2018 18:03:21*

*this sentence has to be in another part lied to the problematic of environemental lack of data or problems.*

AC: this sentence about validation is now in the discussion of the model results.

**Page : 9**

*Auteur : Sujet : Note Date : 04/02/2018 18:01:50*

*the discussion is poor, the main goal of a discussion is to criticize results, methodology and have an objective vision of the research. I think the discussion has to be improved by authors.*

*I don't see the real contribution of the study. Hard work was made , but the text not reflect this work. You have to improve your text.*

AC: We will explain better this issue in the text and we will critically examine it in the discussion. We completely reorganizing the discussion session.

[revised manuscript text omitted]

**Spostato in giù [1]:** The HIRESSS model simulated two past events, one in 2008 and one in 2009, and the validation of the model performance was carried out comparing the results with the landslide regional database.
In particular:
- 31 May 2008: on 28 and 29 May 2008 intense and persistent rainfall was recorded across the Valle d'Aosta region with a total precipitation in the study area of about 250 mm causing flooding, debris flows and rockfalls.
- 28 April 2009: from 26 April to 28 April 2009 heavy rainfall affected the south-eastern part of the Valle d'Aosta region, with the highest precipitation recorded at the Lillianes Granges station of about 268 mm. This precipitation triggered several landslides.

[revised manuscript text omitted]

Most commonly used models to quantify rooted soils strength are based on a Mohr-Coulomb failure criterion for unsaturated soil in which a term representing root reinforcement is added (Eq. 2):

$$\tau = c' + (\mu_a - \mu_w) \tan \varphi_b + (\sigma - \mu_a) \tan \varphi' + c_r$$

(2)

where $\tau$ is the soil-shearing resistance, $c'$ effective cohesion, $\mu_a$ the pore-air pressure, $\mu_w$ the pore-water pressure, $\varphi_b$ the angle describing the increase in shear strength due to an increase in matric suction ($\mu_a - \mu_w$), $\sigma$ the normal stress on the shear plane, $\varphi'$ the effective soil friction angle, and $c_r$ the increase in shear strength due to roots.

Spostato (inserimento) [17]

Spostato (inserimento) [18]

Spostato in su [5]: , roots seem to affect the cohesion parameter only, while the friction angle would be poorly or not at all interested by reinforcement (Waldron and Dakessian, 1981; Gray and Ohashi 1983; Operstein and Frydaman, 2000; Giadrossich et al., 2010).

Spostato in su [6]: The root reinforcement (or root cohesion) can be considered equal to (Eq.

Spostato in su [7]: ): .
$c_r = kT_r(A_r/A)$

[revised manuscript text omitted]

**Spostato (inserimento) [19]**

**Spostato (inserimento) [20]**

**Spostato (inserimento) [21]**

**Spostato (inserimento) [22]**

**Spostato (inserimento) [23]**

**Spostato (inserimento) [24]**

**Spostato (inserimento) [25]**

**Spostato (inserimento) [26]**

[revised manuscript text omitted]

**Pagina 2: [1] Eliminato**               **Teresa Salvatici**                     **26/04/18 10:27:00**

Moreover, in

**Pagina 3: [2] Eliminato**               **Teresa Salvatici**                     **26/04/18 10:27:00**

**3.1 Soil Geotechnical and hydrological characterization**

**The**

**Pagina 5: [3] Eliminato**               **Teresa Salvatici**                     **26/04/18 10:27:00**

:

$$k_s = \frac{Q\left[\sinh^{-1}\left(\frac{h}{r}\right) - \left(\frac{r^2}{h^2}+1\right)^{\frac{1}{2}} + \frac{r}{h}\right]}{2\pi h^2}$$

(1)

where $Q$ is the steady-state rate of water flow from the permeameter into the auger hole, $h$ is the water depth in the borehole (constant), and $r$ is the borehole radius.

**Pagina 6: [4] Eliminato**               **Teresa Salvatici**                     **26/04/18 10:27:00**

**3.2. Evaluation of root reinforcement**

Root reinforcement is due to root tensile strength that is usually greater than the tensile strength of soil. Conversely, soil has a greater strength to compression, therefore the overall effect is a strengthened matrix soil, in which stresses are relocated from sediments to roots (Greenway, 1987). Consequently, the strength of rooted soil results from sediments nature (cohesion and friction angle), root strength and strength of soil-roots bonds (Waldron, 1977; Waldron and Dakessian, 1981; Ennos, 1990). Regarding strength parameters

**Pagina 7: [5] Spostato a pagina 4 (spostamento n. 8)Teresa Salvatici**               **26/04/18 10:27:00**

where Tr is the root failure strength (tensile, frictional, or compressive) of roots per unit area of soil, Ar/A the root area ratio (proportion of area occupied by roots per unit area of soil), k a coefficient dependent on the effective soil friction angle and the orientation of roots. The measure of cr varies with vegetal species, within a single species depends on how plants respond to environmental characteristics and fluctuations.

**Pagina 7: [6] Formattato**               **Teresa Salvatici**                     **26/04/18 10:26:00**

Colore carattere: Testo 1

**Pagina 7: [7] Eliminato**               **Teresa Salvatici**                     **26/04/18 10:27:00**

In view of all that has been mentioned so far, it is necessary to consider the root cohesion in calculating FS and consequently in applying HIRESSS model.

**Pagina 7: [7] Eliminato**               **Teresa Salvatici**                     **26/04/18 10:27:00**

| Pagina 7: [8] Eliminato | Teresa Salvatici | 26/04/18 10:27:00 |

,

| Pagina 7: [8] Eliminato | Teresa Salvatici | 26/04/18 10:27:00 |

,

| Pagina 7: [8] Eliminato | Teresa Salvatici | 26/04/18 10:27:00 |

,

| Pagina 7: [9] Spostato a pagina 3 (spostamento n. 3)Teresa Salvatici | 26/04/18 10:27:00 |

**HIRESSS description**

The physically-based distributed slope stability simulator HIRESSS (Rossi et al., 2013) is a model developed to analyse shallow landslide triggering conditions on large scale at high spatial and temporal resolution using parallel calculation method. Two parts compose the model: hydrological and geotechnical (Rossi et al., 2013). The hydrological part is based on a dynamical input of the rainfall data which are used to calculate the pressure head and provide it to the geotechnical stability model. The hydrological model is initiated as a modelled form of hydraulic diffusivity, using an analytical solution of an approximated form of the Richards equation under the wet condition (Richards, 1931). The equation solution allows us to calculate the pressure head variation ($h$), depending on time ($t$) and depth of the soil ($Z$). The solutions are obtained by imposing some boundary conditions as described by Rossi et al. (2013).

The geotechnical stability model is based on an infinite slope stability model. The model considers the effect of matric suction in unsaturated soils, taking into account the increase in strength and cohesion. The stability of slope at different depths (Z values) is computed since the hydrological model calculates the pressure head at different depths. The variation of soil mass caused by water infiltration on partially saturated soil is also modelled. The original FS equations (Rossi et al., 2013) were modified taking into account the effect of root reinforcement ($c_r$) as an increase of soil cohesion ($c'$) according to the Eq.

| Pagina 7: [10] Spostato a pagina 3 (spostamento n. 4)Teresa Salvatici | 26/04/18 10:27:00 |

:

$$c_{tot} = c' + c_r$$

| Pagina 7: [11] Spostato a pagina 4 (spostamento n. 9)Teresa Salvatici | 26/04/18 10:27:00 |

The new equation of FS at unsaturated conditions is therefore (Eq.

| Pagina 7: [12] Spostato a pagina 4 (spostamento n. 10)Teresa Salvatici | 26/04/18 10:27:00 |

):

$$FS = \frac{\tan\varphi}{\tan\alpha} + \frac{c_{tot}}{\gamma_d y \sin\alpha} + \frac{\gamma_w h \tan\varphi\{[1+(h_b^{-1}|h|)^{\lambda+1}]^{\frac{\lambda}{\lambda+1}}\}^{-1}}{\gamma_d y \sin\alpha}$$ (

| Pagina 7: [13] Spostato a pagina 4 (spostamento n. 11)Teresa Salvatici | 26/04/18 10:27:00 |

where $\varphi$ is the friction angle, $\alpha$ is the slope angle, $\gamma_d$ is the dry soil unit weight, $y$ is the depth, $\gamma_w$ is the water unit weight, $h$ is the pressure head, $h_b$ is the bubbling pressure, and $\lambda$ is the pore size index distribution. In saturated condition the equation of FS (Rossi et al., 2013) becomes (Eq.

| Pagina 7: [14] Spostato a pagina 4 (spostamento n. 12)Teresa Salvatici | 26/04/18 10:27:00 |

):

$$FS = \frac{\tan\varphi}{\tan\alpha} + \frac{c_{tot}}{(\gamma_d(y-h)+\gamma_{sat}h)\sin\alpha} - \frac{\gamma_w h \tan\varphi}{(\gamma_d(y-h)+\gamma_{sat}h)\tan\alpha}$$ (

| Pagina 7: [15] Spostato a pagina 4 (spostamento n. 13)Teresa Salvatici | 26/04/18 10:27:00 |
|---|---|

where $\gamma_{sat}$ is the saturated soil unit weight.

One of the major problems, associated with the deterministic approach employed on a large scale, is the uncertainty of the static input parameters or geotechnical parameters of the soil. The method used for the estimation of parameters spatial variability is the Monte Carlo Simulation. The Monte Carlo simulation achieves a probability distribution of input parameters providing results in terms of slope failure probability

| Pagina 7: [16] Spostato a pagina 4 (spostamento n. 14)Teresa Salvatici | 26/04/18 10:27:00 |
|---|---|

The developed software uses the computational power offered by multicore and multiprocessor hardware, from modern workstations to supercomputing facilities (HPC), to achieve the simulation in reasonable runtimes, compatible with civil protection real time monitoring (Rossi et al. 2013).

| Pagina 7: [17] Eliminato | Teresa Salvatici | 26/04/18 10:27:00 |
|---|---|---|

**3.4 HIRESSS input data**

The HIRESSS model loads spatially distributed data arranged as input raster maps. Therefore, point data and parameters have to be adequately spatially distributed. In this application the spatial resolution was 10 m and 12 raster maps of static input parameters were prepared. These input raster were (Fig. 4): slope gradient; effective cohesion ($c'$); root cohesion ($c_r$); friction angle ($\varphi'$); dry unit weight ($\gamma_d$); soil thickness; hydraulic conductivity ($k_s$); initial soil saturation ($S$); pore size index ($l$); bubbling pressure ($h_s$); effective porosity ($n$); and residual water content ($q_r$).

The slope gradient (Fig. 5a) was calculated from the DEM (Digital Elevation Model). Effective cohesion, friction angle (Fig. 5b), hydraulic conductivity (Fig. 5c), effective porosity (Fig. 5f) and dry unit weight (Fig. 5g), were obtained, spatializing according to lithology, the soil punctual parameters derived from the in situ and laboratory geotechnical tests and analysis carried out as described in sect. 3.1. Soil thickness (Fig. 5e)

| Pagina 7: [18] Spostato a pagina 5 (spostamento n. 15)Teresa Salvatici | 26/04/18 10:27:00 |
|---|---|

was calculated by the GIST model (Catani et al., 2010; Del Soldato et al, 2016). Soil characteristic curves parameters (pore size index, bubbling pressure, and residual water content) were derived from literature values (Rawls et al.,

| Pagina 7: [19] Eliminato | Teresa Salvatici | 26/04/18 10:27:00 |
|---|---|---|

1982) and they are constant in whole area. Root cohesion values (Fig. 5d), at the depth chosen for the physical modelling with HIRESSS, were obtained taking into account vegetational maps (Carta delle serie di vegetazione d'Italia, Italian Ministry of the Environment and Protection of Land and Sea) and values from literature of root cohesion (Bischetti, 2009; Burylo et al.,

| Pagina 7: [20] Spostato a pagina 5 (spostamento n. 16)Teresa Salvatici | 26/04/18 10:27:00 |
|---|---|

2010; Vergani et el., 2013) that were calculated considering the Fiber Bundle Model (Pollen et al., 2004).

| Pagina 7: [21] Eliminato | Teresa Salvatici | 26/04/18 10:27:00 |
|---|---|---|

The initial soil saturation was empirical defined based on antecedent rainfall analysis. Moreover, considering the lithological and land use maps the exposure rock mask (Fig. 5h) was prepared, so that HIRESSS model avoided the simulation on steep rock slopes areas. The parameters are showed in Table 2 for all lithological classes.

**Pagina 7: [22] Spostato a pagina 6 (spostamento n. 17)Teresa Salvatici**     26/04/18 10:27:00

In the study area, the rainfall hourly data from 27 pluviometers were available, therefore it was necessary to spatially distribute them to generate 10x10 m cell size input raster to ensure the correct program operation. The rainfall data were elaborated applying the Thiessen's polygon methodology (Rhynsburger, 1973) modified to take into account the elevation. Thiessen's polygon methodology, in fact, allows us to divide a planar space in some regions, and to assign the regions to the nearest point feature. This approach defines an area around a point, where every location is nearer to this point than to all the others. Thiessen's polygon methodology do not consider the morphology of the area, so the alert Zone B was divided in three catchment areas and the polygons were calculated for each rain gauges considering the reference catchment basin (Fig.

**Pagina 7: [23] Spostato a pagina 6 (spostamento n. 18)Teresa Salvatici**     26/04/18 10:27:00

).

**4 Results**

**Pagina 7: [24] Formattato**     Teresa Salvatici     26/04/18 10:26:00

Colore carattere: Testo 1

**Pagina 7: [25] Eliminato**     Teresa Salvatici     26/04/18 10:27:00

The HIRESSS model provide

**Pagina 7: [25] Eliminato**     Teresa Salvatici     26/04/18 10:27:00

The HIRESSS model provide

**Pagina 7: [25] Eliminato**     Teresa Salvatici     26/04/18 10:27:00

The HIRESSS model provide

**Pagina 7: [25] Eliminato**     Teresa Salvatici     26/04/18 10:27:00

The HIRESSS model provide

**Pagina 7: [25] Eliminato**     Teresa Salvatici     26/04/18 10:27:00

The HIRESSS model provide

**Pagina 7: [26] Spostato a pagina 8 (spostamento n. 19)Teresa Salvatici**     26/04/18 10:27:00

validation is necessary to have information on spatial and temporal location of landslides. In particular, the time of occurrence is very rarely known with hourly precision, and usually landslides are related to a rainstorm, without any more precise information on time of occurrence (Rossi et al., 2013). Concerning the spatial landslides locations, in many cases they are included in the database only as points without any information on the area involved. In our database, provided by the local authorities, landslides are points with information on the day of occurrence.

In general, for both events temporal validation shows that the daily highest probability of occurrence, computed by HIRESSS, correspond with the days with real landslide occurrence and with the most intense precipitation.

The results of the first

**Pagina 7: [27] Formattato**     Teresa Salvatici     26/04/18 10:26:00

Colore carattere: Testo 1

**Pagina 7: [28] Eliminato**     Teresa Salvatici     26/04/18 10:27:00

event (24 - 31 May 2008) are shown in Fig. 7. The failure probability in the whole area is less than 25% for the first four days (from 24 to 27 May 2008) (Fig. 7a

Pagina 7: [29] Spostato a pagina 8 (spostamento n. 20)Teresa Salvatici          26/04/18 10:27:00

). The rainfall intensity increased since 27 May, reaching the highest value on 29 May, when the precipitation value was around 100 mm in the eastern sector of study area.

The HIRESSS model well simulate this passage: the 28 May and 29 May 2008 landslide occurrence probability maps show a considerable increase of the probability of failure with maximum values around 90% at the East of alert Zone B (Fig.

Pagina 7: [30] Spostato a pagina 8 (spostamento n. 21)Teresa Salvatici          26/04/18 10:27:00

 b, c). In the following days rainfall intensity decreases, and also the probability slowly decreases, being anyway still high on 30 May 2008. Landslides reported in the database are dated 30 May and 31 May 2008 (Fig.

Pagina 7: [31] Eliminato                    Teresa Salvatici                    26/04/18 10:27:00

7d).

Concerning the second event (25 - 28 April 2009) landslide occurrence probability is less than 25% for the first two days (25 and 26 April 2009) in the whole area (Fig.

Pagina 7: [32] Formattato                    Teresa Salvatici                    26/04/18 10:26:00

Colore carattere: Testo 1

Pagina 7: [33] Spostato a pagina 8 (spostamento n. 22)Teresa Salvatici          26/04/18 10:27:00

a, b), because of the low rainfall intensity. From 27 April 2009 rainfalls become more intense, especially in the southeast sector of the region, where the cumulated rainfall average was about 151 mm. This event led to many landslides triggered during these days (as reported in the database). Also the probability maps show high values during these days (Fig.

Pagina 7: [34] Spostato a pagina 8 (spostamento n. 23)Teresa Salvatici          26/04/18 10:27:00

The temporal validation was also carried out considering daily cumulative rainfall compared to the landslide failure probability. In particular, a median of landslide occurrence probability was calculated for four pluviometric areas identified by Thiessen's polygons methodology, modified according to limits of river basins, both for the event of May 2008 and for the April 2009 event (Fig.

Pagina 7: [35] Formattato                    Teresa Salvatici                    26/04/18 10:26:00

Colore carattere: Testo 1

Pagina 7: [36] Eliminato                    Teresa Salvatici                    26/04/18 10:27:00

 a, b). As it could be expected, the results show that when the highest rainfall intensity is measured, the highest probability of occurrence is computed for the all areas and for both events.

Pagina 7: [37] Spostato a pagina 8 (spostamento n. 24)Teresa Salvatici          26/04/18 10:27:00

Spatial validation was performed following a pixel by pixel method: this method is the most complex since it consists in comparing the probability of instability of each pixel with the pixels involved in the actual event that occurred. This validation implies a great deal of uncertainty in the results since the reports of landslide events may have errors on the precise spatial location and on the size of the phenomenon. To overcome this problem and taking into account probable errors caused by the actual spatial location in the database, an area of 1 km$^2$ (called influence area) around the point of the landslide were considered in the validation analysis. Inside the influence area, pixels that have the 75% of probability of failure were considered instable.

**Pagina 7: [38] Spostato a pagina 8 (spostamento n. 25)Teresa Salvatici**      **26/04/18 10:27:00**

shows an example of landslide event occurred in the Arnad municipality on 30 May 2008. The model computes a low failure probability on 24 May 2008 and an increase of probability on 30 May 2008. In Fig.

**Pagina 7: [39] Spostato a pagina 8 (spostamento n. 26)Teresa Salvatici**      **26/04/18 10:27:00**

and b it is possible to note that inside the red circle the red and yellow area increase on 30 May with respect to 24 May. In this case, the model is able to identify correctly such movement. To better highlight this validation, Figure 10c shows the number of pixels above 75% of probability calculated by the model, within the circular area of about 1 km$^2$ around the all landslides occurred during the event of 2008. For some of the reported landslide events, the number of pixels above 75% increases on 30 May,2008, only in case of the Champdepraz and Montjovet 2 events the probability does not increase. This may be caused by the low precision of location of the reported landslide, and maybe because some of the real landslides reported are other types of movements (rockfalls, rotational slides) that

**Pagina 7: [40] Formattato**      **Teresa Salvatici**      **26/04/18 10:26:00**

Colore carattere: Testo 1

**Pagina 8: [41] Eliminato**      **Teresa Salvatici**      **26/04/18 10:27:00**

The final aim of the physically-based modelling for landslide prediction is to set-up an early warning system at regional scale based on the model output. The validation of the results performed in the previous section showed that the HIRESSS model performs good results with good prediction capacity both from a spatial and temporal point of view. In this work the HIRESSS model computes the daily probability of occurrence with a spatial resolution of 10 m. In order to become an active and proficient early warning system it is necessary to define a method for the interpretation of the probabilistic results (e.g., definition of probability values corresponding to alert thresholds). Furthermore, in order to have more usable results especially for public administration and civil protection authorities it is necessary to possibly aggregate the model outputs temporally and spatially.

In particular, we selected a spatial aggregation method at the municipality level. Three level of failure probabilities (low, medium and high) are defined based on the expert-judged analysis of the cumulated frequency of the municipality median values of failure probability in the most critical day of the event (e.g., highest rainfall and failure probability). This procedure was done for the two events described in Sect. 4, defining for each of them different failure probability thresholds.

Once defined the three classes of probability, each municipality was classified according to the median value of probability inside its perimeter for each day. The results for the two analysed events are shown in Fig. 11 and Fig. 12. It is worth to notice that for some municipalities with the increase of rainfall intensity there is an increase of failure probabilities values from low (green) to red (high) that can be further translated in alert levels. The validation reported in Table 3 show the number of landslides for each failure class (low, medium high). It is worth noticing that for both events the majority landslides are located in the municipalities with low and medium HIRESSS probability of occurrence.

Figure 11 and Figure 12 are examples of how the model results can be analysed but the validation results are not satisfactory. The results have to be refined and the approach should be tailored to end users needs and requirements, in particular, the following aspects should be taken into account:

> spatial resolution: we have selected the municipality as spatial level of aggregation but also another types of spatial units (e.g., first or second order basins, Rossi et al., 2013) can be taken into account depending on the end-users needs and type of early warning system; temporal resolution: in this work HIRESSS has computed daily failure probabilities. The model is coded anyway to compute FS with different temporal resolutions. In real time applications the model can produce results with different time steps (e.g., six or twelve hours); definition of thresholds: the validation results show that the applied approach based on the analysis of cumulated median values of failure probabilities is not good enough to correctly forecast landslides. Different thresholds should be defined for each spatial unit of the early warning system based on a sound statistical analysis of HIRESSS results. To do a satisfactory analysis is necessary to have a good dataset of past triggered landslides.

Gray, D. H., and Sotir, R. B.: Biotechnical and Soil Bioengineering Slope Stabilization, John Wiley & Sons Inc., 378, 1996.

Greenway, D. R.: Vegetation and slope stability, In Slope Stability, Anderson MG, Richards KS (eds), Wiley, Chichester, 187–230 1987.

Intrieri, E., Gigli, G., Casagli, N., and Nadim, F.: Landslide early warning system: toolbox and general concepts. Nat. Hazard Earth Sys., 13, 85–90, 2013

[Figure]

**Figure 5**

[Figure]

**Figure 11.** Spatial aggregation method at the municipality level for the events of May 2008 according to the value of failure probability.

[Figure]

**Figure 12**. Spatial aggregation method at the municipality level for the events of April 2009 according to the value of failure probability.

---

## Referee Report (RR1)

[referee-annotated manuscript omitted]

---

## Author Response (AR2)

**Answer to the Editors and referees**

Editor comment:

Dear Authors,

the revised version of your manuscript upgrades the previous one. The suggestions followed have increased the comprehension.

However, both reviewers agree on the possibility to further improve this paper. In particular, "Results" and "Discussions" still need a review. The current section "Results" reads more as an explanation of the dataset used in input to the HIRESSS model (as also stated at line 236). On the other hand, lines from 246 to 273 of the "Discussion" basically describe the results obtained by the model. Finally, the "Conclusions" need to be improved and extended, because the first part (lines 293-299) reads more as a summary.

For the above mentioned reasons, I return the paper to you for the last revision. Please, review the manuscript accordingly to these instructions. As a suggestion you could consider to change "Results" into "Geotechnical and hydrological characterization of the case study area". Then, you can move forward to upgrade: "Results", "Discussions" and "Conclusions".

Specific comments:

I would suggest to slightly modify the title in: "Application of a physically-based model to forecast shallow landslides at a regional scale".

Line 285: I guess It should be Figure 11C

> *Answer to the Editor:*
>
> *Dear Editor,*
>
> *We thank you for the careful revision of our manuscript and for the relevant advises. We have revised again the manuscript taking into account your comments and suggestions but also the comments and suggestions of the referees from the first revision and second revision.*
>
> *We would like to highlight anyway that partially the comments of the Editor contradict with the comments of Referee 2 of the first revision, with particular reference to the results of the geotechnical soil characterization.*
>
> *Anyway we have tried to solve the problem as follows (hoping that this solution fits the editor requests):*
>
> - *Results: we have divided the section 4 (Results) into two subsections: 4.1 HIRESSS input data and 4.2: HIRESSS simulation. The section 4.1 includes the results of the input data preparation, that are an essential part of our work and influence the performance of the model. Section 4.2 include the results of the HIRESSS simulation. In this section we have moved the original discussion session since we agree with the editor and the referees that the old discussion section was actually a description of the results.*
> - *Discussion: this section has been completely rewritten and it deals with the major drawbacks of the model application, especially concerning the problems related to the model validation and to the uncertainties of the input data.*
> - *Conclusion: Usually in scientific papers conclusions should summarize the whole work in a concise manner, highlighting the results achieved. For this reason, we have added few sentences at the end of the conclusion, regarding a critical summary of the discussion of the results. We hope that this modification meets the editor requests.*

Editor comment:

Specific comments:

I would suggest to slightly modify the title in: "Application of a physically-based model to forecast shallow landslides at a regional scale".

Line 285: I guess It should be Figure 11C

> *Answer to the Editor:*
>
> *We agree with editor's suggestion and we have modified the title .*
>
> *We have corrected the reference to figure.*

**Anonymous Referee #2**

Please detail the acquired time, spatial resolution and other characteristics of the DEM used in

the model.

> *Answer:*
>
> *We have included in the text the information related to the resolution and the year. We don't have information about other characteristics.*

The revised discussion does not solve my concern raised last time.

Please detail your changes in your response and avoid vague responses to changes you made.

> *Answer:*
>
> *We have modified again the discussion section. In particular we have tried to analyze the major drawbacks of the model application with particular reference to the problem related to the model validation and the uncertainty of the model input data.*

**Anonymous Referee #3**

Dear authors, I read carefully your text.

you have considered the requests of the reviewers and I congratulate you for this work. Nevertheless, as it stands, your text still suffers from some gaps.

If the introduction, the description of the test site and the model correspond to the expected standard, this is not the same thing for the results and the discussion paragraphs. I think there is still confusions between the two parts.

Please find my comments in the pdf file and try to take into account for these.

best regards

> *Answer:*
>
> *We have modified the text according to the referee comments, with particular reference to the Results and Discussion sections. We have taken into account the suggestions of the referee in the pdf file.*